# Image Textualization👁️: An Automatic Framework for Creating Accurate and Detailed Image Descriptions

**Renjie Pi**[1]*, **Jianshu Zhang**[2]*, **Jipeng Zhang**[1], **Rui Pan**[1], **Zhekai Chen**[3], **Tong Zhang**[4]

[1]The Hong Kong University of Science and Technology
[2]Wuhan University,   [3]Zhejiang University
[4]University of Illinois Urbana-Champaign
{rpi,rpan,jzhanggr}@ust.hk, jianshu.zhang@whu.edu.cn, chenzhekai@zju.edu.cn
tongzhang@tongzhang-ml.org

## Abstract

Image description datasets play a crucial role in the advancement of various applications such as image understanding, text-to-image generation, and text-image retrieval. Currently, image description datasets primarily originate from two sources. One source is the scraping of image-text pairs from the web. Despite their abundance, these descriptions are often of low quality and noisy. Another is through human labeling. Datasets such as COCO are generally very short and lack details. Although detailed image descriptions can be annotated by humans, the high annotation cost limits the feasibility. These limitations underscore the need for more efficient and scalable methods to generate accurate and detailed image descriptions. In this paper, we propose an innovative framework termed **Image Textualization (IT)**, which automatically produces high-quality image descriptions by leveraging existing multi-modal large language models (MLLMs) and multiple vision expert models in a collaborative manner, which maximally convert the visual information into text. To address the current lack of benchmarks for detailed descriptions, we propose several benchmarks for comprehensive evaluation, which verifies the quality of image descriptions created by our framework. Furthermore, we show that LLaVA-7B, benefiting from fine-tuning on IT-curated descriptions, acquire improved capability to generate richer image descriptions, substantially increasing the length and detail of their output with less hallucination.

## 1 Introduction

In recent years, multi-modal large language models (MLLMs) have witnessed significant progresses. Such models start to reach super-human performances in a variety of areas, such as image understanding [2, 15, 34, 41, 80], text-to-image generation [13, 54, 55, 57, 58] and text-image retrieval [33, 53, 69, 77]. One of the primary reasons for these successes is the training data, which consists of image-description pairs. Recent studies highlight that the quality of image descriptions is crucial for MLLM performance. For example, Yin et al. [74] note that low-quality descriptions often cause hallucinations in image understanding tasks, while Betker et al. [4] show that detailed descriptions with richer visual concepts significantly enhance generation models' performance. Thus,

---

* Equal Contribution. Code and data are available at the following links:
https://github.com/sterzhang/image-textualization/
https://huggingface.co/datasets/Sterzhang/image-textualization/.
The code and data are released under MIT and apache2.0 licenses, respectively.

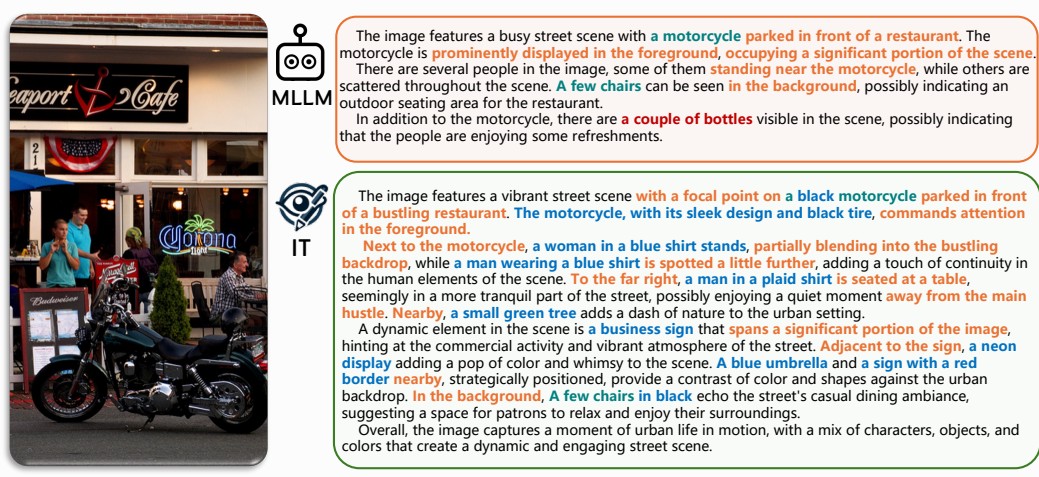

Figure 1: Visualization of our Image Textualization. Compared with the MLLM-generated description, our description incorporates more visual details and significantly less hallucinations. The shared details, newly added details, hallucinations, and positional descriptions are all marked with different colors.

curating high-quality image description datasets is essential for improving various downstream applications.

A high-quality image description should convey the same information as the corresponding image, effectively textualizing the visual content. However, current datasets often fall short. They mainly come from two sources: web-scraped image-text pairs [8, 60, 61], which are large-scale but low-quality and noisy, and human-labeled datasets [31, 39, 52], which lack in-depth details and are costly to produce. Consequently, a significant gap remains between the information in an image and its textual description.

To address the shortcomings of existing image description datasets, recent advancements in MLLMs have shown remarkable potential for generating descriptions. Dalle-3 [4] has made an early attempt to train text-to-image diffusion models using descriptions produced via MLLMs, which shows improved quality of the generated images. However, MLLMs possess several weaknesses, such as the well known visual hallucination problem [51, 74, 75] and lack of fine-grained details [12]. Even the most powerful MLLMs, such as GPT4-V [44], exhibit these weaknesses. Therefore, relying solely on MLLMs to generate datasets still has significant limitations.

Meanwhile, we notice remarkable progress in areas of computer vision, such as object detection [42, 72, 73], dense captioning [43, 67] and instance segmentation [30]. Compared with MLLMs that are trained with low-resolution images (336x336 by default setting of CLIP [53]) and image-level annotations, these vision expert models are trained with high-resolution images and fine-grained object-level annotations specifically catering for perception tasks, which makes them capable of identifying detailed content. However, such models generally do not possess holistic understanding capabilities, so constructing descriptions solely depending on such models is not practical. Consequently, an intriguing thought arises: Can we combine the understanding capability of MLLMs with the perception power of vision experts to generate high-quality descriptions that are both rich in details and free from hallucinations.

Building upon the above intuition, in this paper, we propose **Image Textualization(IT)**, a framework for automatically creating high-quality image descriptions. Specifically, our framework consists of three phases: 1) *Holistic Textualization*: We leverage the MLLM to create the Reference Description, which, despite lacking details and containing hallucinations, provides the basic structure not only for the visual information but also for the linguistic expression. 2) *Visual Detail Texturalization*: Then, we resort to the powerful perception capabilities of vision expert models to extract fine-grained object-level information that is converted into text format. This phase extracts multiple details from the image-side and identifies hallucinations contained in the Reference Description. 3) *Textualized Recaptioning*. Finally, we leverage LLMs' superior understanding and reasoning capabilities to

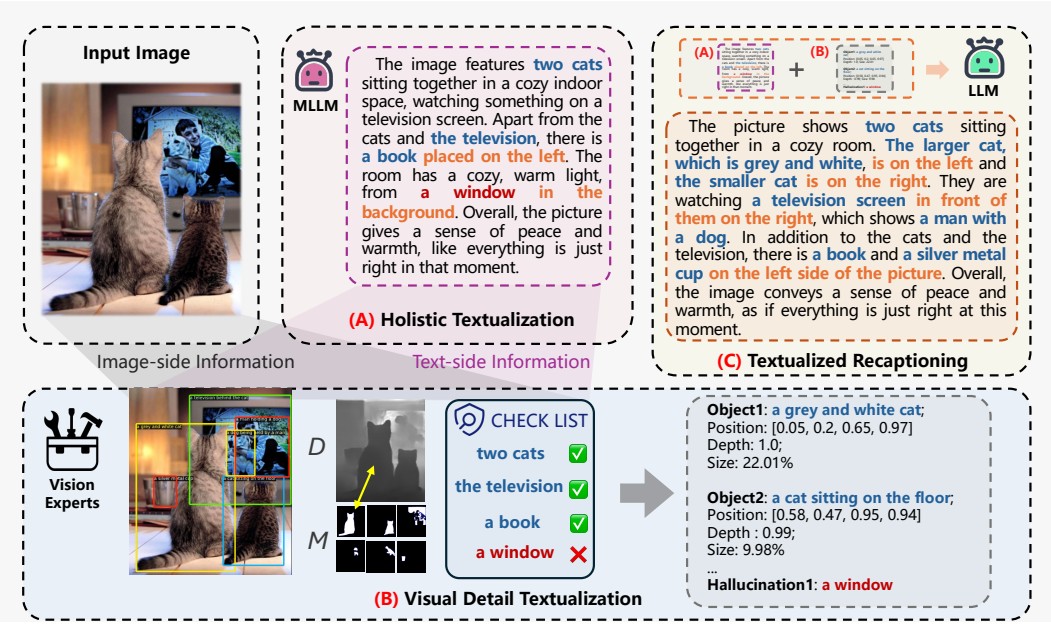

Figure 2: The framework of **Image Textualization (IT)**, which consists of three phases: (A) **Holistic Textualization** (Sec. 3.1) utilizes a MLLM to generate a "*Reference Description*" that provides a basic structure; (B) **Visual Detail Textualization** (Sec. 3.2) identifies the hallucinations and captures details in the image via a variety of vision experts, then transforms them to text format. (C) **Textualized Recaptioning** (Sec. 3.3), which leverages LLM and textualized results from (A) and (B) to re-generate the image captions that are both rich in details and free from hallucination.

produce accurate and detailed descriptions based on the textualized information from the first two phases, allowing the LLMs to describe the image without "seeing" it. This approach avoids the weaknesses of MLLM-based recaptioning. As shown in Figure 1, our IT framework is able to create image descriptions that are richer in details and free from hallucination.

For a comprehensive evaluation of our framework, we first construct three benchmarks, namely DID-Bench, D2I-Bench and LIN-Bench, which evaluate the description quality from multiple aspects. Then, we conduct a series of experiments to validate the quality of IT-generated descriptions on the proposed benchmarks. Afterward, we verify that fine-tuning MLLMs with our generated data enhances their capabilities. Lastly, we perform linguistic evaluations and provide statistical analysis of our released dataset.

To summarize, we make the following contributions in this paper:

- We propose **Image Textualization**, a framework that automatically generates detailed image descriptions without human intervention, leveraging the multimodal understanding of MLLMs, the fine-grained perception of visual experts, and the reasoning power of LLMs.

- We create evaluation benchmarks and conduct extensive experiments to validate the effectiveness of our framework. The results demonstrate that the generated image descriptions accurately capture rich visual details.

- Using our Image Textualization framework, we curate a large-scale high-quality image description dataset termed **IT-170K**. To facilitate future research, we release all the source code and our generated dataset to the community.

## 2 Related Work

**Image Description Datasets**    Image-text paired description datasets are valuable assets for a variety of downstream tasks, such as image understanding [2, 15, 41, 66, 80], text-to-image generation [13,

54, 55, 57, 58] and text-image retrieval [33, 53, 69, 77]. Image description datasets primarily curated from three sources: 1) web-scraped image-text pairs, such as CC3M [61], CC12M [8] and LAION [60], which are large-scale, but often contain low quality and noisy descriptions. 2) human-labeled description datasets, such as COCO [39], Flickr30k [52] and Visual Genome [31], which typically have limited quantity and often feature short and incomplete captions due to the costly annotation process. The lack of high-quality image description datasets may cause unsatisfactory performance or out of domain problem [9–11, 78]. To address this gap, in this paper, we propose a framework that automatically generates high-quality, detailed image descriptions without human intervention.

**Multi-Modal Large Language Model.** In recent years, great advancements have been made in the development of large language models (LLMs) [3, 6, 14, 25, 45, 59, 62, 65]. These advancements have greatly elevated the capabilities of language understanding and generation, showcasing super-human proficiency across diverse tasks. Concurrently, the success of LLMs has inspired explorations into the incorporation of visual modality into LLM, leading to the emergence of multi-modal large language models (MLLMs) [2, 15, 15, 16, 20, 21, 34, 41, 44, 48–50, 63, 80]. These models have demonstrated remarkable abilities in engaging in dialogue based on visual inputs and generating image descriptions containing rich details. Despite the success of MLLMs, their inherent weaknesses such as low image resolution and insufficient training data, leads to problems such as incomplete description and object hallucination [35, 51, 64, 74, 75]. Therefore, directly generating image descriptions with MLLMs remains problematic.

**Vision Expert Models** There is a variety of sub-fields in computer vision that specialize on different tasks. Object detection aims at localizing objects in images [7, 17, 22, 40, 56, 70, 71, 81]. Recently, open-vocabulary object detection has achieved great progress at localizing objects based on the semantics of text queries [24, 36, 42, 72, 73]. Dense captioning aims to produce short descriptions for each object present in the input images [26, 67, 73]. Prompt-based segmentation models enable producing segmentation mask for objects in the image based on the input prompts, which could be in the form of point or bounding box [27, 30, 82]. Depth estimation enables the prediction of distance between objects and the camera [29, 68]. In this paper, we harness the capabilities of the vision experts to provide object-level information for constructing high-quality image descriptions.

## 3 Method

**Image Textualization** automatically produces high-quality image descriptions. Given an image, the powerful MLLM first produces a template description capturing the holistic image content. Then, a variety of vision expert models collaborate to extract detailed object information, which may be missing from the template description. Finally, we harness the powerful LLMs to re-generate the description based on the holistic information and fine-grained details. As shown in figure 2, our framework is divided into three phases, which we will elaborate in the following sections.

### 3.1 Phase 1: Holistic Textualization

Current state-of-the-art MLLMs [15, 41, 44] excel at producing image descriptions that contain richer information and contextual understanding compared with those generated by conventional captioning models [18, 23, 34, 66]. Therefore, we first leverage MLLM-generated descriptions to textualize the holistic content of the image, as shown in Figure 2 Phase(A). Despite weaknesses such as hallucinations and lack of details, this description can serve as a basic template with a relatively good structure for describing the image. Hereafter, we refer to this description as the *"Reference Description"*.

This Reference Description serves two key purposes. Firstly, in terms of visual information, it includes the main objects present in the image and the contextual information of the scene. These elements act as "anchors" that guide the incorporation of more details in the subsequent phases. Secondly, from a linguistic expression perspective, the inherent understanding and logical capabilities of MLLMs help to form well-organized descriptions. For example, a Reference Description typically includes an overall description of the image, followed by details about the main objects, then concludes with a summarizing sentence. Compared to traditional captioning models, this kind of descriptions are more logically structured and naturally expressed, which are crucial factors for the quality of descriptions.

## 3.2 Phase 2: Visual Detail Textualization

Reference descriptions generated in the first phase generally lack in visual details and contain hallucinations. In this phase, we utilize vision expert models to extract information from both the image and reference description. From the image, we capture more visual details, and from the reference description, we identify the hallucinated contents. Finally, we textualize the fine-grained visual information and hallucinated objects, as shown in Figure 2 Phase(B) rightmost grey box.

### 3.2.1 Hallucination Detection

As outlined in Algorithm 1, to identify hallucinations existed in the reference description, we first extract object entities (object nouns and phrases) from it. Here, we leverage the strong instruction following ability of the Large Language Model (LLM). Specifically, we carefully design an entity-extraction prompt and manually annotate in-context learning examples to improve instruction following of the LLM. Afterward, we utilize an open-set object detector (e.g., Grounding Dino [42]) to verify each of these extracted entity phrases against objects in the image. Any hallucinated object phrases, which are not found in the image, are tagged as "Hallucination" for removal in the later phase.

---

**Algorithm 1** Hallucination Detection

**Require:** An input image $\mathcal{I}$, a description $\mathcal{T}$, a large language model *LLM*, an openset object detector $\rho(\cdot)$.
1: Initialize *Hallucination* as Empty
2: $\mathcal{P} \leftarrow$ **LLM**$(\mathcal{T})$     *# extract object phrases*
3: **for** each phrase $p_i \in \mathcal{P}$ **do**
4:    $tag_i \leftarrow \rho(I, p_i)$  *# tag hallucination*
5:    Append $tag_i$ to *Hallucination*
6: **end for**
7: **Output:** *Hallucination*

---

### 3.2.2 Fine-Grained Objects Annotation

**Dense Caption Generation.** To identify objects that are potentially missing in the original description, we resort to a dense captioner (DC) [67], which not only provides accurate bounding boxes indicating object locations, but also associate them with basic attributes, such as object type, shape and color. Compared with object detectors that only predict the object category (e.g., cat), DC provides a more detailed description such as "a grey and white cat"; compared with conventional captioning models that only predicts image-level captions, DC is able to predict descriptions for all the visible objects in the image. These appealing properties make DC a suitable choice for maximizing the textualization of objects' information, which is beneficial for the subsequent recaptioning phase.

---

**Algorithm 2** Fine-grained Object Annotation

**Require:** An input image $\mathcal{I}$ with size $H \times W$, dense caption model *DC*, a segment anything model *SAM*, a monocular depth estimator parameterized by $\mathcal{F}(\cdot)$.
1: Initialize *FinegrainedInfo* as Empty
2: $\mathcal{B}, \mathcal{P} \leftarrow DC(\mathcal{I})$    *# get object boxes and phrases*
3: $\mathcal{M} \leftarrow SAM(\mathcal{I}, \mathcal{B})$         *# obtain object masks*
4: $\mathcal{D} \leftarrow \mathcal{F}(\mathcal{I})$             *# obtain image depth map*
5: **for** each object mask and phrase $m_i, p_i \in [\mathcal{M}, \mathcal{P}]$ **do**
6:    $d_i \leftarrow \frac{\sum(m_i \odot \mathcal{D})}{\sum m_i}$              *# obtain object depth*
7:    $s_i \leftarrow \frac{\sum m_i}{H \times W}$                *# obtain object size*
8:    Append $\{p_i, d_i, s_i\}$ to *FinegrainedInfo*
9: **end for**
10: **Output:** *FinegrainedInfo*

---

**Spatial Information Collection.** Now, we have obtained the dense captions of various objects in the image along with their bounding box coordinates. However, the textualized information still falls short compared to the original image's information. The most critical reasons for this is that the current textualization can only convey the relative left-right relationships of objects on a 2D plane, and can lead to mistakes in recaptioning. For example, consider an image with a car in the background and a person in the foreground. Their bounding boxes might have very close coordinates. If we move to the Recaptioning phase with just this information, the LLM might use its logical capabilities to inaccurately describe the scene as "a person standing next to a car". This happens because the current textualization fails to capture the 3D spatial context, such as depth (which indicates the front-back relationships of objects), which are crucial for an accurate and comprehensive image description.

As shown in Algorithm 2, we derive this depth information by first obtaining a distance map using a monocular depth prediction model, where the value for each pixel indicates its distance from the camera. Then, we generate object segmentation masks with SAM [30] using the object bounding boxes generated by the dense captioner, which gives the exact pixels of the objects. Finally, the object depth is obtained by averaging the depth values within its corresponding segmentation mask.

Table 1: Evaluation of image descriptions on *DID-Bench*. Descriptions generated by our IT outperform the ones generated by MLLMs by a significant margin across different metrics.

| GroundTruth | Description | BLEU-1 | BLEU-2 | BLEU-3 | BLEU-4 | CIDEr | METEOR | ROUGE | SPICE | WMD |
|---|---|---|---|---|---|---|---|---|---|---|
| GT-{LLaVA} | {LLaVA} | 12.90 | 8.64 | 5.80 | 4.09 | 0.00 | 12.84 | 22.69 | 23.08 | 43.82 |
| | IT-{LLaVA} | **25.71** | **17.53** | **12.06** | **8.68** | **2.34** | **17.09** | **26.10** | **26.10** | **46.49** |
| | {GPT4-V} | 29.05 | 15.23 | 7.64 | 4.15 | 1.93 | 15.92 | 20.06 | 19.84 | 42.79 |
| | IT-{GPT4-V} | **36.20** | **19.97** | **10.75** | **6.23** | **7.64** | **18.56** | **21.34** | **22.35** | **43.81** |
| GT-{GPT4-V} | {LLaVA} | 9.80 | 5.16 | 2.54 | 1.35 | 0.00 | 9.83 | 15.93 | 13.75 | 37.93 |
| | IT-{LLaVA} | **21.86** | **12.17** | **6.67** | **3.94** | **1.18** | **13.80** | **18.74** | **17.86** | **40.12** |
| | {GPT4-V} | 45.26 | 38.77 | 34.42 | 31.18 | 6.08 | 26.63 | 50.85 | 52.21 | 58.52 |
| | IT-{GPT4-V} | **57.38** | **48.73** | **43.02** | **38.89** | **36.67** | **30.89** | **54.36** | **55.20** | **61.23** |

In this way, the textualized information can convey the distance between the object and the camera, effectively transforming the 2D dense captions into 3D ones.

In this process, we remark the two important factors: 1) **Utilizing Pixel Masks for Size Calculation**: One naive way to calculate object size is using the bounding boxes. However, bounding boxes only provide two opposite corners, which can be inaccurate for irregularly shaped objects (e.g., a long stick). If we solely rely on bounding box's coverage as the object's size, it can lead to severe overestimation. Hence, it is crucial to use pixel-wise masks to accurately represent the object's size. 2) **Normalization**: We normalize the bounding box coordinates and depth scores to ensure that the values are relative, which facilitates the LLMs to adapt to different images during recaptioning phase.

### 3.3   Phase 3: Textualized Recaptioning.

In this phase, we utilize the comprehensive information gathered from the previous phases that has been transformed into a textual representation, to reconstruct the image description via an LLM. The inclusion of both a holistic description and extensive object-level information enables the LLM to accurately interpret the entire image and its constituent objects, eliminating the need for direct visual input. We demonstrate the effectiveness of our approach in the appendix, where we carefully design prompts and provide few-shot examples to guide the LLM's generation process. This ensures that the LLM can effectively incorporate novel objects while minimizing the presence of hallucinated content.

## 4   Experiments

**Overview**   Due to lack of standard evaluation benchmarks for long image descriptions, we first propose DID-Bench, D2I-Bench and LIN-Bench for comprehensively evaluating detailed descriptions(Sec. 4.1). Then, we conduct a series of experiments to validate the effectiveness of our Image Textualization that can generate high-quality descriptions(Sec. 4.2). Afterward, we verify that training MLLMs with the data generated by our framework enhances their capabilities (Sec. 4.3). Lastly, we perform linguistic evaluations and provide statistical analysis of our released dataset (Sec. 4.4).

### 4.1   Benchmarks and Evaluations

**DID-Bench**   Detailed Image Description Bench (*DID-Bench*) contains 200 samples via the following steps: 1) First, we utilize MLLMs to generate Reference Descriptions. To avoid the bias introduced by different MLLMs' output habits, we employ GPT4-V for 100 samples and LLaVA for the remaining 100 samples when generating the Reference Descriptions. 2) Then, we manually check the correctness of these descriptions, add missing details, and remove hallucinated content to establish the human-labeled ground truth descriptions. We later refer to the partition using LLaVA Reference Descriptions to get ground truth as GT-{LLaVA}, and the other partition as GT-{GPT4-V}.

We adopt reference-based metrics for image descriptions including BLEU [47], ROUGE-L [38], METEOR [32], SPICE [1] and WMD [28]. These metrics evaluate various aspects of the generated descriptions, such as n-gram overlap, recall, precision, semantic content, and overall similarity to human-labeled descriptions. The details of these metrics are introduced in the appendix.

**D2I-Bench**   We propose Description-to-Image-Bench (D2I-Bench) to evaluate the completeness of image information captured by descriptions. Firstly, we feed the descriptions to a pre-trained text-to-image model (e.g., PixArt [13]) and obtain the generated images. Then, we extract the image

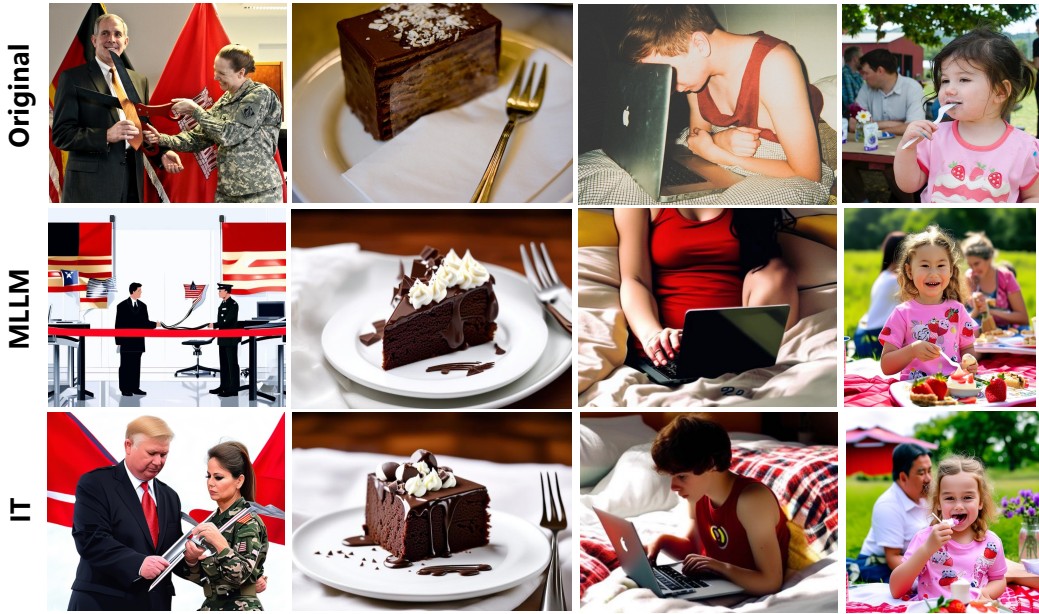

Figure 3: D2I-Bench visualization. IT-generated descriptions capture more fine-grained image details, which leads to generated images more similar to the original images.

embeddings for both the original image and the generated image using the image encoder from a pre-trained CLIP [53]. Finally, we calculate the cosine similarities between the image embeddings. A higher similarity score indicates that the description effectively captures the image details, resulting in generated images that closely resemble the originals.

**LIN-Bench** To fully evaluate the readability and linguistic features of descriptions, we propose Linguistic Bench (LIN-Bench), which adopt metrics such as ARI, FK, and SMOG. ARI tends to be higher if there are more words with many characters in the sentence, while FK and SMOG place more emphasis on the number of multi-syllable words. More detailed descriptions tend to achieve higher values for these metrics.

**POPE** POPE benchmark [37] evaluates the level of hallucination suffered by the MLLM, which comprises of questions regarding the existence of objects in the image, and associated with short answers such as "yes" or "no".

## 4.2 Image Description Quality Evaluation

**DID-Bench Results** To verify the effectiveness of our Image Textualization annotation framework for generating detailed and accurate image descriptions, we compare the quality of descriptions generated by our Image Textualization with the ones directly produced by the MLLMs. As shown in Table 1, we observe significant gain across all the metrics for different MLLMs and ground-truth annotations. Interestingly, we observe that the evaluation results of IT-generated descriptions also dependent on the MLLM used in the holistic textualization phase. This is because the evaluation metrics not only account for the visual correctness, but also the styles of the descriptions, such as pronouns and prepositions. Therefore, to exclude the impact of language bias, it is crucial to conduct evaluation using GT annotations with different styles.

**D2I-Bench Results** As shown in table 4, the descriptions generated by our IT framework results in images that have higher similarity scores with the original images than COCO's descriptions and MLLM-generated descriptions. In figure 3, we provide qualitative examples to show the IT-generated

descriptions leads to images that bear closer resemblance as the originals. These results demonstrate the effectiveness of our framework for accurately capturing the details of the image content.

Table 2: Evaluation on POPE and Lin-bench. LLaVA trained with IT-generated data produces richer image descriptions and demonstrates alleviated hallucination.

| Tuning Data | Num | POPE | | | | LIN-Bench | | | |
| --- | --- | --- | --- | --- | --- | --- | --- | --- | --- |
| | | Adv | Rand | Popular | Average | ARI | FK | SMOG | Average |
| / | - | 79.13 | 85.70 | 88.93 | 84.59 | 8.80 | 8.48 | 10.93 | 9.40 |
| {LLaVA} | 10k | 79.60 | 86.16 | 89.56 | 85.11 | 8.77 | 8.45 | 10.91 | 9.38 |
| IT-{LLaVA} | | **81.37** | **87.40** | **90.63** | **86.47** | **9.99** | **9.48** | **11.3** | **10.26** |
| {GPT4-V} | 10k | 83.46 | **88.03** | 90.23 | 87.24 | 8.78 | 8.53 | 11.14 | 9.51 |
| IT-{GPT4-V} | | **83.60** | 88.00 | **90.47** | **87.36** | **10.03** | **10.89** | **11.89** | **10.94** |
| {GPT4-V} | 50k | 81.96 | 88.03 | 90.13 | 86.71 | 9.57 | 8.89 | 11.08 | 9.85 |
| IT-{GPT4-V} | | **83.30** | **88.20** | **90.80** | **87.43** | **10.47** | **9.65** | **11.62** | **10.58** |

## 4.3 MLLM Tuning Evaluation

**DID-Bench Results**   In Table 3, we compare the quality of image descriptions generated by 1) the original LLaVA-7B model; 2) LLaVA fine-tuned with MLLM-generated descriptions and 3) LLaVA fine-tuned with descriptions produced by Image Textualization. We observe that the MLLM fine-tuned using IT-curated dataset consistently outperforms other baselines across all metrics and GT annotations. We observe the following phenomena: 1) the scores on GT-LLaVA is mostly higher than that of GT-GPT4V, which is response style of LLaVA; 2) For each GT split, IT-tuned LLaVA outperforms the baseline and the MLLM-tuned LLaVA by a large margin; 3) from the evaluation on combined GT, we observe IT-LLaVA's effectiveness approaches that of GPT4-V, while IT-GPT4-V still surpass all counterparts significantly. This indicates that Image Textualization has the potential to close the gap between the capability of different MLLMs.

**POPE and LIN-Bench Results**   On the left side of Table 2, we evaluate the hallucination of MLLMs tuned with different image description data. We observe that tuning with IT-generated descriptions leads to the most significant alleviation of hallucination. On the right side, we also show the results on LIN-Bench, which demonstrates that tuning with IT-generated descriptions results in most gain in producing descriptions containing richer details.

## 4.4 Linguistic-based Evaluation

**Statistical Analysis**   We summarize the statistics of image descriptions generated by Image Textualization and MLLM baselines, respectively. In Table 5, we show that the IT-generated descriptions contain more words and sentences. In the figure on the right, we show the counts for different types of words, which demonstrates that the IT-generated descriptions contain richer words such as nouns, verbs and adjectives, indicating these descriptions contain denser information.

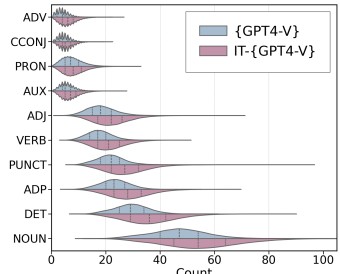

**LIN-Bench Results**   As demonstrated in Table 6. Compared with MLLM-generated descriptions, IT-generated descriptions results in higher scores across all metrics, suggesting that these descriptions contain richer details.

## 5 Prompt Design

Prompt design plays a crucial role in our proposed framework, which enables the collaboration between different expert models and results in more accurate and detailed image descriptions. In figure 4, we compare the recaption results without fine-grained object annotation and in-context examples and observe that object annotation leads to more detailed and accurate description, while in-context

Table 3: Evaluation of LLaVA's ability to generate image descriptions on *DID*. We find that LLaVA, when tuned with IT-generated descriptions, achieves significantly better performance compared with tuning using MLLM-generated descriptions and the baseline (Tuning Data is /).

| GroundTruth | Tuning Data(10k) | BLEU-1 | BLEU-2 | BLEU-3 | BLEU-4 | METEOR | ROUGE | SPICE | WMD |
|---|---|---|---|---|---|---|---|---|---|
| GT-{LLaVA} | / | 12.90 | 8.64 | 5.80 | 4.09 | 12.84 | 22.69 | 23.08 | 43.82 |
| | {LLaVA} | 11.61 | 7.11 | 4.21 | 2.61 | 11.61 | 20.61 | 18.41 | 42.36 |
| | IT-{LLaVA} | **23.59** | **15.65** | **10.39** | **7.21** | **16.34** | **24.81** | **26.76** | **45.62** |
| | {GPT4-V} | 30.62 | 18.03 | 10.48 | 6.47 | 16.57 | 23.88 | 20.73 | 44.87 |
| | IT-{GPT4-V} | **37.88** | **22.59** | **13.24** | **8.12** | **17.38** | **24.52** | **21.32** | **44.89** |
| GT-{GPT4-V} | / | 9.80 | 5.16 | 2.54 | 1.35 | 9.83 | 15.93 | 13.75 | 37.93 |
| | {LLaVA} | 10.22 | 5.54 | 2.76 | 1.44 | 10.05 | 16.78 | 14.34 | 38.20 |
| | IT-{LLaVA} | **23.47** | **12.74** | **6.49** | **3.48** | **12.77** | **19.04** | **16.27** | **39.23** |
| | {GPT4-V} | 27.24 | 15.09 | 8.15 | 4.68 | 15.33 | 21.34 | 18.71 | 42.08 |
| | IT-{GPT4-V} | **35.28** | **19.57** | **10.37** | **5.77** | **16.79** | **21.93** | **19.23** | **42.41** |
| GT-{LLaVA} & GT-{GPT4-V} | / | 11.35 | 6.90 | 4.17 | 2.72 | 11.33 | 19.31 | 18.41 | 40.87 |
| | {LLaVA} | 10.92 | 6.33 | 3.49 | 2.03 | 10.83 | 18.69 | 16.38 | 40.28 |
| | IT-{LLaVA} | **23.78** | **14.85** | **9.36** | **6.31** | **15.45** | **22.42** | **21.98** | **43.30** |
| | {GPT4-V} | 28.93 | 16.56 | 9.32 | 5.57 | 15.95 | 22.61 | 19.72 | 43.48 |
| | IT-{GPT4-V} | **46.79** | **34.35** | **26.89** | **22.56** | **24.72** | **37.85** | **38.77** | **52.52** |

Table 4: D2I-Bench Results.

| Description | CLIP-score | DINO-score |
|---|---|---|
| COCO | 72.24 | 77.84 |
| {LLaVA} | 73.44 | 80.39 |
| IT-{LLaVA} | **74.27** | **81.20** |
| {GPT4V} | 76.49 | 82.80 |
| IT-{GPT-4V} | **77.10** | **83.71** |

Table 5: Description statistics.

| Description | #Word | #Sentence |
|---|---|---|
| {LLaVA} | 92.57 | 5.08 |
| IT-{LLaVA} | **131.61** | **6.17** |
| {GPT4-V} | 159.93 | 8.96 |
| IT-{GPT4-V} | **193.33** | **9.82** |

Table 6: LIN-bench Evaluation.

| Description | ARI | FK | SMOG | Avg |
|---|---|---|---|---|
| {LLaVA} | 9.34 | 8.87 | 11.25 | 9.74 |
| IT-{LLaVA} | **11.02** | **10.23** | **12.04** | **10.75** |
| {GPT4-V} | 9.82 | 9.02 | 11.22 | 10.05 |
| IT-{GPT4-V} | **10.75** | **9.83** | **11.74** | **10.64** |

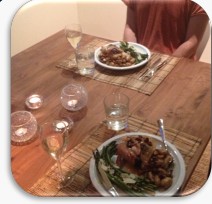

**w/o annotations + shots**

This image captures a cozy dinner setting …
Beside each plate … A wine glass sits on the table alongside a clear glass of water. **A knife and a fork** are neatly arranged on the table. Additionally, **a silver knife** rests on one of the plates, emphasizing the meticulous arrangement of the table elements.
The table setting is completed with white candles housed in glass holders, casting a warm and inviting glow over the scene. The presence of **a person sitting at the table** adds a personal touch, enhancing the atmosphere of intimacy and comfort. The overall composition of the image conveys …

**w/o shots**

This image captures a cozy dinner setting …
Beside each plate, there are multiple glasses - some filled with water, such as **the one** positioned at **[0.46, 0.48, 0.59, 0.67]** with a relative size proportion of **1.99%** and **another** at **[0.25, 0.43, 0.39, 0.62]** with a relative size proportion of **1.49%**, and others with white wine, like **the one** at **[0.27, 0.13, 0.36, 0.34]** with a relative size proportion of **0.9%**. The table setting is completed with white candles housed in glass holders, casting a warm and inviting glow over the scene, with **one candle** positioned at **[0.0, 0.56, 0.14, 0.74]** with a relative size proportion of **2.09%**.
The image exudes … On the table, there are also utensils, including **a knife and fork** at **[0.64, 0.3, 0.75, 0.41]** with a relative size proportion of **0.23%** and a silver butter knife at **[0.43, 0.81, 0.56, 1.0]** with a relative size proportion of **0.59%**. There is **a person sitting at the table** at **[0.55, 0.0, 1.0, 0.32]** with a relative size proportion of **10.78%**. The overall composition of the image conveys …

**ours**

This image captures a cozy dinner setting …
Beside each plate, there are multiple glasses - some filled with water, such as **the one** positioned near **the center-right of the table**, and **another** near **the upper-left side**, and others with white wine, like **the one** positioned toward **the upper-left corner** of the table. The table setting is completed with white candles housed in glass holders, casting a warm and inviting glow over the scene, with **one candle** positioned toward **the lower-left side** of image on the table.
The image exudes … On the table, there are also utensils, including **a knife and fork** positioned near **the upper center-right** and **a silver butter knife** near **the bottom-center**. There is **a person sitting at the table, occupying a large portion of the upper right corner of the image**. The overall composition of the image conveys …

**Reference Description**

This image captures a cozy dinner setting. At the center of the frame is a rectangular, wooden table adorned with brown, woven placemats. On each placemat, there is a white plate filled with a delicious meal consisting of a protein, green beans, and mushrooms.
Beside each plate, there are multiple glasses - some filled with water and others with white wine. The table setting is completed with white candles housed in glass holders, casting a warm and inviting glow over the scene.
The image exudes an atmosphere of intimacy and comfort, perfect for a quiet dinner for two. The meticulous arrangement of the table elements suggests careful preparation and attention to detail. The overall composition of the image conveys a sense of harmony and balance, making it a delightful visual treat.

Figure 4: Comparison with results generated without using fine-grained annotation and in-context examples. We find that the few-shot examples and the detailed information provided by the visual experts are crucial for high-quality image descriptions.

examples effectively prevent the description to contain exact values of fine-grained information. We incorporate more detailed prompts for other phases in the appendix.

# 6 Limitation

While our framework demonstrates promising robustness against hallucinations and enhances downstream performance, several limitations warrant consideration. Firstly, despite claims of mitigating misinformation, some provided examples exhibit inaccuracies—for instance due to the performance bottleneck of vision expert models.

Additionally, our approach relies exclusively on model-based techniques, which introduces inherent limitations related to model diversity. Specifically, models like GPT-4V exhibit a strong positivity bias, often describing images in overly favorable terms such as "cozy" and "beautiful." This lack of diversity can result in biased data, as the training and evaluation processes predominantly reflect these inherent model tendencies.

Furthermore, the potential misuse of image description generation technologies poses significant risks, including the creation of false news or misleading information. Our current work does not address these ethical concerns, nor does it explore the implications related to privacy and bias in depth. These issues are critical, given the powerful capabilities of image description models and their impact on information dissemination. Incorporation with advanced data selection approaches is promising [5, 19, 46, 76, 79].

Lastly, while we acknowledge the limitations associated with using smaller LLaVa models, we have not thoroughly examined the broader societal impacts of our work. Moreover, our reliance on evaluation metrics such as BLEU and ROUGE, which are known to be noisy and sometimes unreliable, underscores the need for more robust assessment methods. These metrics may not fully capture the qualitative aspects of image descriptions, potentially affecting the validity of our evaluations.

In summary, future work should address these limitations by incorporating diverse model architectures, mitigating biases, exploring ethical implications, and employing more reliable evaluation metrics to enhance the robustness and societal relevance of our framework.

# 7 Conclusion

In conclusion, this paper addresses the limitations of existing image description datasets and proposes an innovative framework, Image Textualization (IT), to generate detailed and accurate image descriptions. The framework leverages the power of multimodal large language models (MLLMs) and multiple vision expert models in a collaborative manner. Through extensive experiments for image understanding and generation tasks, we validate the high quality of the descriptions generated by the framework. We hope our work provides inspiration for the design of more efficient and scalable methods to generate detailed and accurate image descriptions.

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

In this appendix, we first provide the detailed prompts for object entity extraction via LLM, which was adopted in phase 2 for hallucination identification. Then, we illustrate the prompts and in-context examples for textualized recaptioning. Next, we showcase more qualitative comparisons between IT-generated and MLLM-generated image descriptions. Finally, we point out the limitation of our work, which may be addressed in future works.

## A  Detailed Prompt Design for Object Entity Extraction in Phase 2

In table 7, we demonstrate the detailed prompt that guides the LLM to perform entity extraction from the template description. Specifically, we first indicate the role of the LLM. Then we emphasize the things to remember during the extraction process: 1) only extract the objects that certainly exists in the image; 2) avoid extracting the background objects or intangible items; 3) the response should follow a certain format to facilitate parsing. Next, we provide the LLM with human-annotated in-context examples to enable better instruction following ability. Lastly, we provide the LLM with the new template image description to perform entity extraction. In table 8, we showcase the in-context examples provided to LLM for entity extraction.

---

**Extract Prompt**

**###TASK DESCRIPTION###**

Your are a helpful entities extractor. Please help me to extract the OBJECTS mentioned in a description about an image.

**###THINGS TO REMEMBER###**

1. Only extract the descriptions of objects that are described with certainty. For example, in the sentence "there's a white car parked, perhaps belonging to one of the hotel guests," the "hotel guests" part is included within "perhaps," indicating uncertainty. Therefore, you only need extract "a white car" that is described with certainty.
2. Avoid extracting abstract or non-specific entities (such as "cozy atmosphere", "excitement", "sky view")!!!
3. Your response should strictly start with "%%%RESPONSE%%%:" , following this format: "%%%RESPONSE%%%: obj1. obj2. obj3...."

**###IN-CONTEXT EXAMPLES###**
Here are some examples for your reference.
<In-Context Examples>

**###TASK###**
%%%DESCRIPTION%%%: <Description>
%%%RESPONSE%%%:

---

Table 7: The prompt for extracting entities in the description, <In-Context Examples> is the place-holder for several in-context examples that illustrated in Table 8. <Description> will be replaced by the description to be modified.

## B  Prompt for Textualized Recaptioning in Phase 3

In table 9, we demonstrate the prompt for textualized recaptioning. We first inform the LLM of its role as a recaptioner. Then, we provide the detailed explanation for the object-level spatial information, i.e., Relative spatial position, relative depth from lens and relative size of the objects. Next, we emphasize the following points: 1) avoiding duplication when incorporating new objects into the description, 2) the photographic characteristics mentioned in the Original Description should be preserved; 3) the exact values of the spatial information (e.g., bounding boxes) should not be incorporated into the description. Afterwards, we provide human-annotated in-context examples to enhance the annotation quality.

In table 11, we showcase one of the in-context examples provided to the LLM for textualized re-captioning.

## In-Context Examples for the Extract Prompt

**Example 1:**

%%%DESCRIPTION%%%: This image captures a cozy Christmas scene. At the center of the image is a bed adorned with a red satin bedspread with gold tassels and a gold pillow. The bed is nestled within red curtains that have gold trim, adding to the festive atmosphere.
On the bed, there are four teddy bears, two of which are brown and the other two are white. They are arranged in a pile, creating a sense of warmth and comfort. Adding to this feeling is a white cat that is peacefully sleeping next to the teddy bears.
The bed is further decorated with red and gold Christmas ornaments, enhancing the holiday spirit. To the right of the bed stands a Christmas tree adorned with gold ornaments, while on the left side of the bed, there are two red candlesticks.
Overall, this image exudes a warm and festive atmosphere, perfect for the holiday season.

%%%RESPONSE%%%: red satin bedspread. gold tassels. gold pillow. red curtains. gold trim. four teddy bears. white cat. red and gold Christmas ornaments. Christmas tree. gold ornaments. two red candlesticks.

---

**Example 2:**

%%%DESCRIPTION%%%: The image depicts a lively scene in an ornately decorated room. At the center of the image stands a man with a long white beard, wearing a tall, cylindrical black hat with a flat top. He is dressed in a white shirt paired with a green tie and carries a black backpack.
The room is bustling with people, some of whom are wearing green shirts, adding to the vibrant atmosphere. The room itself is richly adorned, featuring a gold ceiling and red columns that exude an air of grandeur.
The man, despite being in the midst of the crowd, stands out due to his unique attire and commanding presence. The people are visible on all sides of him, suggesting that he is at the heart of this gathering.

%%%RESPONSE%%%: man with a long white beard. tall cylindrical black hat. white shirt. green tie. black backpack. people wearing green shirts. gold ceiling. red columns.

---

**Example 3:**

%%%DESCRIPTION%%%: This image captures a charming scene in a bedroom. The main subject is a black cat standing on its hind legs on a gray nightstand. The cat, full of curiosity, is reaching its paw into a white cup adorned with blue and green designs.
The nightstand hosts a few other items: a beige lamp, a white tissue box with blue flowers, and a pink jar of Vaseline. Each item is neatly arranged on the nightstand, creating a harmonious tableau. In the background, you can see a bed covered with a white comforter and adorned with a yellow pillow, adding a pop of color to the scene. The precise location of these objects and their relative positions contribute to the overall composition of the image, creating a snapshot of everyday life with a hint of feline mischief.

%%%RESPONSE%%%: black cat. gray nightstand. white cup with blue and green designs. beige lamp. white tissue box with blue flowers. pink jar of Vaseline. white comforter. yellow pillow.

Table 8: In-context examples for object entity extraction.

## C    Qualitative Comparison between IT-generated and MLLM-generate Image descriptions

We provide more qualitative comparisons between MLLM-generated and IT-generated image descriptions in table 12, table 13 and table 14. We observe that compared with MLLM-generated descriptions, IT-generated ones incorporate more comprehensive visual details, and also demonstrate less hallucination.

## Recaptioning Prompt

### ###TASK DESCRIPTION###
You are a helpful language assitant. Imagine visualizing an image based on its description. Now, your task is to make the Original Description more detailed. You'll need to use the subsequent provided Objects, along with the corresponding extra information of ***Relative Spatial Positioning***, ***Relative Distance from the Lens***, and ***Relative Size Proportion in Images (Percentage)***, to better assist you in adding these Objects to the original description.

### ###EXTRA INFORMATION EXPLANATION###
1. ***Relative Spatial Positioning***: It uses a normalized coordinate system where both x (horizontal) and y (vertical) axes range from 0 to 1. The x-coordinate starts at 0 on the image's left edge and increases to 1 towards the right edge. Similarly, the y-coordinate starts at 0 at the top edge and increases to 1 towards the bottom. This system uses four coordinates to define the corners of a rectangle within the image: [x1, y1, x2, y2], representing the top-left and bottom-right corners of the rectangle, respectively. For instance, a positioning of [0.00, 0.00, 0.50, 0.50] means the object's top-left corner is at (0.00, 0.00) and its bottom-right corner is at (0.50, 0.50), placing the object in the upper left quarter of the image. Similarly, [0.50, 0.00, 1.00, 0.50] positions the object in the upper right quarter, with corners at (0.50, 0.00) and (1.00, 0.50). A positioning of [0.00, 0.50, 0.50, 1.00] places the object in the bottom left quarter, with corners at (0.00, 0.50) and (0.50, 1.00), while [0.50, 0.50, 1.00, 1.00] positions it in the bottom right quarter, with corners at (0.50, 0.50) and (1.00, 1.00). Moreover, by comparing these coordinates, you can determine the relative positions of objects. For example, an object with positioning [0.20, 0.20, 0.40, 0.40] is to the left of another with [0.30, 0.30, 0.50, 0.50].
2. ***Relative Distance from the Lens***: It measures how far objects are from the camera within the image. The closer the value is to 1, the nearer the object is to the camera, placing it in the foreground. Conversely, a value close to 0 indicates that the object is far from the camera, situated in the background. If two objects share the same Object Distance from the Lens value, it suggests they are at the same depth or layer within the image. For instance, if one object has a value of 0.12 and another is at 0.05, these small differences suggest that both objects are likely in the background, even if not exactly at the same depth but relatively close. This helps in understanding the spatial arrangement of elements in the image and how they relate to the viewer's perspective.
3. ***Relative Size Proportion in Images (Percentage)***: It can tell you the approximate proportion of the object within the image. If the proportion is particularly small or large, it should be emphasized. If the proportion is moderate, it does not need to be highlighted.

### ###THINGS TO REMEMBER###
1. Through the extra information of different Objects, some Objects may represent the same thing. When adding Objects to the Original Description, it is important to avoid duplication.
2. The photographic characteristics mentioned in the Original Description should be preserved, such as sizes, locations, camera angles, depths;
3. The ***Relative Spatial Positioning***, ***Relative Distance from the Lens***, and ***Relative Size Proportion in Images (Percentage)*** of each Object need to be emphasized. However, these details must be expressed without directly using specific values of the extra information, you must convey this information naturally through logical reasoning.

### ###IN-CONTEXT EXAMPLES###
*[Chain of thought is placed within a pair of "@@@" (remember only in the Examples will you be provided with a chain of thoughts to help you understand; in the actual task, these will not be given to you.)]*
<IN-CONTEXT EXAMPLES>

### ###TASK###
*[**You only need to provide the modified description directly after "%%%Your Modified Description:%%%" that I provided**.]*

%%%The Original Description:%%%

<FINE-GRAINED OBJECTS' ANNOTATIONS>

%%%Your Modified Description:%%%

Table 9: Recaptioning Prompt for the last phase "Textualized Recaptioning". <IN-CONTEXT EXAMPLES> is the placeholder for several in-context examples that illustrated in Table 10, 11. <FINE-GRAINED OBJECTS' ANNOTATIONS> will be replaced by the visual detailed textualization created during the second phase.

## In-Context Example of Recaptioning Prompt

**%%%The Original Description:%%%** At the center of the frame is a **black Rolex clock** mounted on a **black pole**. The clock has a **white face** with **black hands**, indicating the time.Behind the clock, there are a **brown tree trunk** with a rough texture and a traffic light. Further back, there's a **white building** with a **red tile roof**, possibly a hotel as indicated by the sign that reads "**HOTEL**".In front of the building, there's a **white car** parked, perhaps belonging to one of the hotel guests. Beyond this immediate scene, there's a street with a **white crosswalk**, where a bus is parked.

---

Hallucinations: a traffic light; a bus
@@@
1. In The Original Description, there is a sentence "Behind the clock, there are a brown tree trunk with a rough texture and a traffic light". Since "a traffic light" is a hallucination, it should be removed, then the sentence should be "Behind the clock, there are a brown tree trunk with a rough texture".
2. In The Original Description, "there's a street with a **white crosswalk**, where a bus is parked" should be modified to "there's a street with a **white crosswalk**", since "a bus" is a hallucination that needs to be deleted.
@@@
Object1: the clock face is white
Relative Spatial Positioning: [0.23, 0.06, 0.55, 0.31]
Distance from the Lens: 1.0
Relative Size Proportion in Images (Percentage): 5.58
@@@
1. The original description already mentions "The clock has a **white face** with **black hands**". Since the object detail is already included, no modification is necessary for this object.
@@@
...

Object5: a person walking on the sidewalk
Relative Spatial Positioning: [0.98, 0.57, 1.0, 0.6]
Distance from the Lens: 0.0
Relative Size Proportion in Images (Percentage): 0.05
@@@
1. Based on the Object Relative Spatial Positioning [0.98, 0.57, 1.0, 0.6], it can be inferred that "a person walking on the sidewalk" is positioned in the middle of the right side of the image.
2. Because the original description includes the line "Beyond this immediate scene, there's a street with a white crosswalk", we can logically associate "a person walking on the sidewalk" with this statement as a supplemental detail.
3. Furthermore, because the Object Distance from the Lens is 0.0, and the Relative Size Proportion of Objects in Images is 0.05, it indicates that "a person walking on the sidewalk" is positioned very far from the camera and is very small. Therefore, we can modify the original sentence "Beyond this immediate scene, there's a street with a white crosswalk." to "Beyond this immediate scene, there's a street with a white crosswalk. At a distance from the camera, you can see a person walking on the sidewalk, occupying a very small part of the entire image."
@@@

---

**%%%Your Modified Description:%%%** At the center of the frame is a **black Rolex clock** mounted on a **black pole**. The clock has a **white face** with **black hands**, indicating the time.the clock, there's a **brown tree trunk** with a rough texture. Further back, there's a **white building** with a **red tile roof**, possibly a hotel as indicated by the sign that reads "**HOTEL**".front of the building, there's a white car parked, perhaps belonging to one of the hotel guests. Closer to the camera, there is a parked silver car near a stop sign. Beyond this immediate scene, there's a street with a white crosswalk. At a distance from the camera, you can see a person walking on the sidewalk, occupying a very small part of the entire image.

Table 10: One of the in-context example of the recaptioning prompt.

**In-Context Example of Recaptioning Prompt**

%%%**The Original Description:**%%% In the heart of a winter wonderland, a lone skier, clad in a vibrant orange jacket, carves their way down a pristine, snow-covered slope.

---

Hallucinations:
@@@
1. There are no hallucinations.
@@@
Object1: a man in a red jacket
Relative Spatial Positioning: [0.8, 0.76, 0.87, 0.93]
Distance from the Lens: 0.96
Relative Size Proportion in Images (Percentage): 0.5
@@@
1. In the original description, it mentions "a lone skier, clad in a vibrant orange jacket." Due to the similarity between an orange jacket and a red jacket, and based on "a lone skier," indicating only one person, the detail "a man in a red jacket" can be disregarded.
@@@
Object2: a backpack on the skier's back
Relative Spatial Positioning: [0.81, 0.76, 0.83, 0.81]
Distance from the Lens: 0.92
Relative Size Proportion in Images (Percentage): 0.04
@@@
1. Because the Object Relative Spatial Positioning and Object Distance from the Lens of this object are very similar to Object1's, "a backpack on the skier's back" most likely refers to "a lone skier, clad in a vibrant orange jacket." Therefore, the detail of "a backpack" should be added, resulting in "a lone skier, clad in a vibrant orange jacket, with a backpack on his back."
@@@

---

%%%**Your Modified Description:**%%% In the heart of a winter wonderland, a lone skier with a backpack on his back, clad in a vibrant orange jacket, carves their way down a pristine, snow-covered slope.

Table 11: One of the in-context example of the recaptioning prompt.

**MLLM-generated Description vs IT-Description**

**Original Description:**

This is a well-organized kitchen with a clean, modern aesthetic. The kitchen features a white countertop against a white wall, creating a bright and airy atmosphere.

On the countertop, you can see a variety of appliances and items. There's a sleek coffee machine, ready to brew a fresh cup. Next to it is a paper towel holder, standing tall and at the ready for any spills. A vase adds a touch of elegance to the space, while a blender suggests the possibility of smoothies or soups being made here. Various bottles and jars are also present, perhaps containing spices or cooking ingredients.

The objects are mostly colored in shades of white, black, and silver, complementing the modern look of the kitchen. However, there are also pops of color with some yellow and teal objects, adding a bit of cheerfulness to the space.

Above the countertop, shelves hold additional items. The arrangement is neat and everything appears to have its own place.

In the foreground of the image is a wooden chair, perhaps providing a spot to sit while waiting for that coffee to brew or the blender to finish its job. In the background, there's a window letting in natural light.

Overall, this kitchen is not only functional with its various appliances and ample storage, but also stylish with its color scheme and neat arrangement.

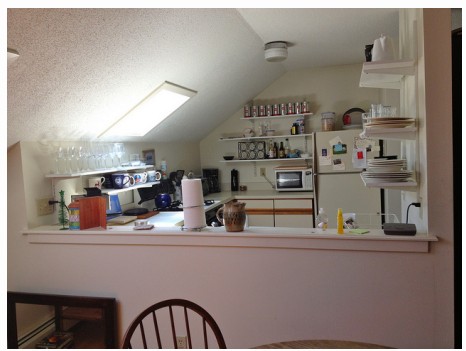

**Modified Description:**

This is a well-organized kitchen with a clean, modern aesthetic, featuring a white countertop against a white wall, creating a bright and airy atmosphere.

On the countertop, an array of appliances and items can be seen. There's a sleek coffee machine ready to brew a fresh cup of coffee, positioned next to a sturdy paper towel holder. A vase adds a touch of elegance, while a blender hints at potential smoothies or soups. Various bottles and jars are also present, suggesting a collection of spices and cooking ingredients. Next to the countertop, there's a white refrigerator in the kitchen, occupying a significant portion of the space. It stands out with its pristine white finish. Nearby, a white microwave oven sits, ready for quick heating or cooking tasks.

Above the countertop, shelves are meticulously arranged with an assortment of items that blend practicality and aesthetic appeal. Here, bottles of wine are aligned next to several clear and stemmed wine glasses that catch the light beautifully. A blue mug and multiple white coffee mugs, additional clear glasses provide contrast and color to the arrangement. Also, there's a variety of cans including a blue, a white can and a can of beer, alongside a silver can, contributing to the diversity of the kitchen's storage contents. The kitchen's color palette mostly features shades of white, black, and silver, complementing its modern design. However, pops of color from the blue pot on the stove, and a small green plant near the cooking area add vibrancy and life to the space.

In the foreground, a brown wooden table and a wooden chair with rods offer a cozy spot to sit and savor the culinary delights from this well-appointed kitchen. Nearby, a brown ceramic cup suggests recent use or a forthcoming coffee break The background is softly illuminated by natural light streaming through a window, enhancing the overall ambiance and providing a welcoming atmosphere.

Overall, this kitchen not only functions effectively with its various appliances and ample storage but also stands out with its stylish color scheme and orderly arrangement, making it a visually appealing and highly practical space for culinary activities.

Table 12: Visualization of the original description and the modified description.

## MLLM-generated Description vs IT-Description

**Original Description:**

The image depicts a well-organized kitchen scene. The primary focus is a white electric range and oven. The range features four burners and a digital clock on the back panel. The oven below has a window and a handle on its door.

To the right of the range, a set of knives is neatly arranged, hanging on the wall. Adjacent to it on the counter, there's a cookbook open, perhaps suggesting someone is preparing to cook a meal. On the left side of the range, there's a black telephone mounted on the wall, adding a touch of vintage charm to the setting.

The backdrop for these items is a white tile backsplash, providing a clean and bright atmosphere to the kitchen.

The image also contains an OCR text reading "03 22 2010 13 21", but its relevance to the kitchen scene is not clear from the image alone. Overall, this kitchen scene suggests preparation for cooking, with tools and appliances neatly arranged and ready for use.

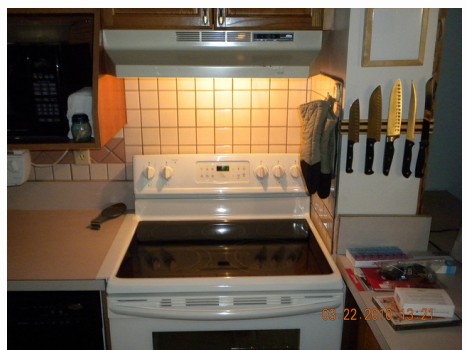

---

**Modified Description:**

The image depicts a well-organized kitchen scene. The primary focus is a white electric range and oven. The range features four burners and a digital clock on the back panel. The oven below has a window and a handle on its door. To the right of the range, a series of knives with black handles are prominently displayed, some hanging on a magnetic strip and others in a cabinet, suggesting a readiness for culinary activities.

To the right of the oven, the counter hosts a variety of items: an open cookbook, suggesting an ongoing meal preparation, and several books, one of which is a prominent red book. Nearby, a box of white paper is also placed, indicating the kitchen's multipurpose use.

On the left side of the range, there is a spoon. A black microwave is conveniently positioned on the counter and a jar finds its place on a shelf, contributing to the organized look.

The backdrop features a white tile backsplash, enhancing the kitchen's bright and clean atmosphere. The entire setting is punctuated by the presence of utensils and books, each placed thoughtfully to suggest both utility and comfort in the kitchen space.

The image also contains an OCR text reading "03 22 2010 13 21", but its relevance to the kitchen scene is not clear from the image alone.

Overall, this image portrays a kitchen ready for both cooking and everyday family life, with every item neatly arranged.

Table 13: Visualization of the original description and the modified description.

## MLLM-generated Description vs IT-Description

**Original Description:**

The image presents a striking contrast between two distinct bathroom designs.

On the left, we see a bathroom that exudes a cool, calming aura with its blue upper wall and green lower wall. The toilet, in harmony with the lower wall, is also green. A white towel, neatly hung on the wall, adds a touch of cleanliness and order to the space.

On the right, we are introduced to a bathroom that embraces a warmer, more vibrant color scheme. The wall is painted a lively shade of orange and is adorned with a tree decal, adding an element of nature to the indoor space. A black cabinet and a white toilet form the main fixtures in this bathroom. Above the cabinet, a black shelf holds jars and a plant, further enhancing the natural theme. A window opens up to a view of trees, connecting the indoors with the outdoors and creating an open, airy atmosphere. In both bathrooms, despite their contrasting designs and color schemes, there is an evident emphasis on cleanliness, orderliness, and harmony with nature. Each design offers its own unique interpretation of comfort and style.

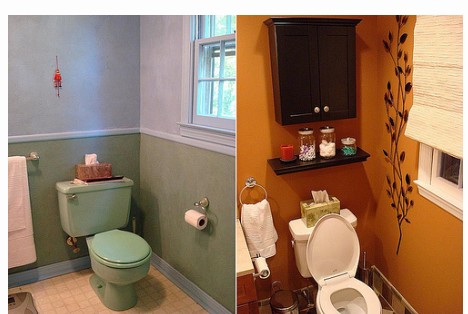

**Modified Description:**

The image vividly captures two contrasting bathroom interiors, each with its unique design elements and functional fixtures that reflect distinct aesthetics of comfort and style.

On the left side, the bathroom showcases a soothing atmosphere with its upper walls painted in a tranquil blue and the lower walls in a refreshing green. The focal point is a light green toilet, perfectly blending with the lower wall, and equipped with a light green toilet seat, emphasizing the cool color palette. Nearby, a towel on a rack hints at meticulous upkeep and functionality. Additional touches include a box of tissues and a roll of toilet paper, conveniently placed, ensuring everything is within reach. A red and white bag hangs from a hook, injecting a pop of color. A window in the bathroom opens to an outdoor view, bridging the interior with the natural world outside. Additionally, a silver trash can, subtly positioned near the toilet, further speaks to the thoughtful arrangement of this space.

Transitioning to the right, the bathroom adopts a warmer and more vibrant decor with its lively yellow walls adorned with a tree decal, creating an inviting natural ambiance. A white toilet anchors this space, complemented by a dark brown cabinet mounted on the wall, which hosts a clear jar with a lid, jars with white napkins, totally three jars on a shelf, all contributing to the room's warm and cozy feel. A small white towel hangs on a towel rack above a silver metal bathroom trash can, combining practicality with style. A window dressed with white window blinds offers a view of the outdoors, linking the room with nature and allowing natural light to brighten the space. Both bathrooms, through their deliberate use of colors, fixtures, and decorative elements, emphasize cleanliness, orderliness, and a harmonious integration with nature. The thoughtful placement of everyday items like toiletries on a shelf and a silver trash can ensures that both practicality and aesthetics are beautifully balanced, offering a comforting and stylish environment.

Table 14: Visualization of the original description and the modified description.

