# OpenReview forum: "Image Textualization: An Automatic Framework for Generating Rich and Detailed Image Descriptions"
_NeurIPS.cc/2024/Datasets_and_Benchmarks_Track — NeurIPS 2024 Track Datasets and Benchmarks Poster_

### Official Review · Reviewer_MZYA · 2024-07-22
**Paper review**

**Rating:** 6
**Confidence:** 5
**Clarity:** The paper is easy to read.

**Review:**

This paper is easy to understand and the authors, by providing the prompts they are using as well as the caption they generated made an important contribution for the community. However, I have some concerns over the quality of the submission itself. One of the contribution of this work is to release the generated dataset, however the authors did not provide a datasheet (or data card) which I believe is an important issue for this conference track. In addition authors mentioned in the main paper that they provide additional experimental details in the appendix, however those details are not in the appendix. Lastly, the authors did not cite LLaVa and most of the citations does not even included the conferences and publications information. Most of the bibliography is mostly authors list and paper titles.

**Strengths:**

- Captions available on hugging face.
- Code available on github.
- The authors provided the model prompts they are using.

**Additional Feedback:**

Overall this is a good engineering and data release contribution. But I believe that the quality of the paper is not up to neurips requirements. The lack of data card or datasheet missing is concerning.

**Correctness:**

The claims made in the submission seems to be correct. The methods and the available caption are constructed in a sound way.

**Documentation:**

No, there isn't sufficient detail since the authors did not provide a datasheet or data card.

**Ethics:**

No.

**Limitations:**

The authors only mentioned the limitations of using small LLaVa models, they did not discussed any potential negative societal impact. Since the authors rely a lot of score such as BLEU/ROUGE, it would have been important to remind the reader that such metrics are noisy and not very reliable.

**Opportunities For Improvement:**

- The authors failed to cite LLaVa even if they are using this model a lot. It is also not clear if the authors are using LLaVa 1.5 or LLaVa 1.6.
- Lack of ablation over the LLaVa model size. Would be interested to know wether there is an improvement or not when using LLaVa 1.6 34B.
- Authors do not provide a datasheet.
- Lack of errors bars. Not clear how randomness is impacting the D2I bench results. Would have appreciate a qualitative assessment over a dozen of generated images.
- The authors did not discussed the societal impact of their work.
- Lack of experimental details in the appendix. The authors wrote in the main paper that "details of these metrics are introduced in the appendix." However, no details on the metrics are available in the appendix.
- I am wondering what would happen if the authors applied their method a second time (like if doing some loop over the last LLM generated caption could help).
- Not sure how much results on hallucinations are significant on POPE. It's true the IT seems to improve a bit the score, but I am worried that there might still be a lot of hallucinations happening.

**Relation To Prior Work:**

Yes.

**Summary And Contributions:**

This paper leverage VLMs to generate captions. Then, to improve the generated captions and avoid hallucinations, the authors leverage a mixture of vision expert models to detect which objects the models is actually hallucinating or missing. Then, they leverage a LLM that take the first generated caption along the corrections given by the expert models, to generate a new, higher quality and reliable caption. Their experiment setup focuses mostly on LLaVa and GPT4-V.

---

> ### Author Rebuttal · Authors · 2024-08-15
>
> Dear Reviewer **MZYA**,
>
> We are grateful for your constructive comments and insightful feedback. We are happy to know that you find our work to have **good engineering and data release contribution**. In the following, we wish to clarify and address the potential concerns.
>
> **(1) LLaVA citation and LLaVA version**: We apologize for the oversight. We will add the citation to LLaVA. The version we used is LLaVA 1.5.
>
> **(2) Generalization on different model sizes**: We perform additional training on LLaVA 13B using our generated descriptions, and demonstrate the results in the table below. However, limited computational resources prevent us from experimenting with LLaVA 34B.
>
> The performance comparison on DID-Bench and POPE are shown below, respectively:
> | Description | BLEU-1 |  BLEU-2 | BLEU-3 | BLEU-4 | CIDEr | METEOR | ROUGE | SPICE | WMD |
> |-|-|-|-|-|-|-|-|-|-|
> | LLaVA-13B | 21.70 | 11.10 | 5.16 | 2.53 | 1.67 | 13.51 | 18.46 | 16.36 | 40.92 |
> | IT-LLaVA-13B | **25.85** | **13.23** | **6.26** | **3.36** | **2.95** | **14.66** | **19.21** | **17.28** | **41.54** |
>
> | Model | Random | Popular | Adversarial |
> |-|-|-|-|
> | {LLaVA-13B} | 84.73 | 81.70 | 77.27 |
> | IT-{LLaVA-13B} | **86.90** | **84.60** | **80.63** |
>
> We observe significant performance improvement after tuning LLaVA-13B with our dataset. Unfortunately, due to limited computational resources, we are not able to conduct training experiments with LLaVA 34B.
>
> Nevertheless, we are pleased and honored that a recent SOTA MLLM incorporated our dataset in their training [1]. The size of their models ranges from 0.5B to 72B. The superior performances of their models confirm that our dataset is effective for models with various parameter counts.
>
>
> **(3) Missing datasheet**: We updated the datasheet on huggingface.
>
> **(4) Error Bars**: Thank you very much for the suggestion. We conduct 5 inferences with different random seeds, then provide the results with added standard deviation below:
>
> | Description | CLIP score | DINOV2 Score |
> |-----------------|----------------|----------------------|
> | COCO | 71.87 (2.13) | 77.15 (1.96) |
> | {LLaVA} | 73.26 (1.24)| 79.64  (1.17)|
> | IT-{LLaVA} | 74.11 (0.91) | 80.85 (1.10)|
> | {GPT-4V} | 76.28 (1.13)| 81.39 (1.24)|
> | IT-{GPT-4V} | 77.31 (0.87)| 83.25  (0.92)|
>
> We will update the error bars in the next version of our paper.
>
> **(5) Qualitative assessment over a dozen of images**: We provide additional examples in the provided PDF. IT-generated descriptions produce images that capture more details of the original images. We will add those examples in the next version.
>
> **(6) Societal Impact**: Image description is crucial in many areas, such as training multimodal foundation models [2], multimodal large language models for image understanding [3], and text-to-image generation models [4]. These fields have notable societal impacts. For example, text-to-image models inspire designers and artists, while multimodal language models can serve as personal assistants processing both text and images. We believe our framework will aid in producing high-quality image description datasets, which is essential for strong model performances. We will elaborate on this in the updated paper.
>
> **(7) Experimental Details**: We apologize for omitting the experimental details in the appendix. Here are the hyperparameters used in our IT framework:
>
> - **Phase 1 (Holistic Textualization)**:
>   - Temperature: 0.2
>   - Top_p: 0.95
>
> - **Phase 3 (Textualized Recaptioning)**:
>   - Temperature: 0.75
>   - Top_p: 0.95
>
> - **For LLaVA fine-tuning experiments**:
>   -  Lora rank: 128
>   - Learning rate: 2 × 10^-5
>   - Batch size per GPU: 8
>   - Training: 2 epochs on 8 A40 GPUs with 40GB memory each.
>
> We will include the missing details in the next version.
>
>
> **(8) Apply a second time**: Great point. In our experiments, we generated descriptions using our framework, fine-tuned the MLLM with them, and then repeated the process with the fine-tuned MLLM as the reference description generator. However, this bootstrapping approach showed no noticeable improvement. We believe this is because the vision experts consistently provide the same image information, and the missing content is added to the template descriptions after the first execution, leaving no new information for subsequent rounds.
>
> **(9) Hallucination results**: Note that our paper does not focus on completely solving the hallucination suffered by MLLMs, since there are many other factors leading to hallucination other than data quality, such as pretraining bias [5] and insufficient image resolution [6]. The improvement in POPE serves as an indicator that our generated image descriptions are accurate, since even though the descriptions produced by our framework contain much richer visual details, the trained MLLM still suffers from less hallucination.
>
> **(9) Noisy evaluation**: To comprehensively evaluate our generated image descriptions, we focus on three aspects: 1) using established rule-based metrics, 2) training MLLMs using our dataset, and 3) analyzing diffusion model-generated images based on our descriptions. We believe our evaluation provides insights into the quality of our dataset. However, we agree that a perfect evaluation for detailed image descriptions remain an open problem and requires more future research, and we will include this in the  limitation section.
>
> We also wish to note that a recent SOTA MLLM incorporated our dataset for training [1], which verifies our dataset's helpfulness for the training of general purpose MLLMs and the acknowledgement from the community.
>
> [1] LLaVA-OneVision: Easy Visual Task Transfer
>
> [2] Learning Transferable Visual Models From Natural Language Supervision
>
> [3] Visual Instruction Tuning
>
> [4] High-Resolution Image Synthesis with Latent Diffusion Models
>
> [5] Mitigating Object Hallucinations in Large Vision-Language Models through Visual Contrastive Decoding
>
> [6] Mini-Gemini: Mining the Potential of Multi-modality Vision Language Models

---

> > ### Comment · Reviewer_MZYA · 2024-08-19
> >
> > Thank you for answering my concerns and for running additional experiments! I updated my score. But, do not forget to add the datasheet (Following the format from "Datasheets for Datasets, Gebru et al, 2018") in the appendix of your paper.

---

> > ### Author Rebuttal · Authors · 2024-08-20
> >
> > Dear Reviewer MZYA,
> >
> > Thank you very much for your positive feedback and for updating the score! We have revised the data sheet in a format similar to the one you referenced, and we have gained substantial insights from your suggestions.
> >
> > We would like to inquire if there are any remaining concerns or additional revisions needed. According to the NeurIPS dataset and benchmark standards, a score of 6 indicates borderline acceptance, which differs from the main track where it represents weak acceptance. We are eager to further enhance our manuscript during the rebuttal period and would greatly appreciate any additional guidance you can provide.
> >
> > Thank you again for your valuable time and assistance!

---

### Official Review · Reviewer_g394 · 2024-07-24

**Rating:** 6
**Confidence:** 3
**Correctness:** yes
**Clarity:** yes

**Review:**

will add soon

**Strengths:**

Strengths:

1. The IT framework innovatively solves the challenges in image description generation by integrating the semantic understanding of MLLM and the detail recognition capability of visual expert models, and ensures that the generated descriptions are both comprehensive and detailed through the three-phase approach of overall textualisation, visual detail textualisation and textual re-description.

2. The authors' proposed evaluation benchmarks, such as DID-Bench, D2I-Bench and LIN-Bench, comprehensively cover multiple dimensions of description quality, providing a reliable evaluation tool for subsequent research, and verifying the advantages of the IT framework in generating high-quality image descriptions by comparing it with existing MLLM-generated descriptions.

3. Through the textualisation of visual details and the fine-grained visual information extracted by the visual expert model, the IT framework is able to effectively identify and reduce the illusory phenomena in the MLLM-generated descriptions, and improves the authenticity and reliability of the descriptions; meanwhile, the descriptions generated by the IT framework are richer in details, and are closer to the human descriptions of the images.

**Additional Feedback:**

N.a.

**Documentation:**

yes

**Limitations:**

Questions:

1. Does the article consider computational cost and resource accessibility issues, especially for research teams with limited resources?

2. How well does the IT framework generalise to different types of image datasets?

3. Do the evaluation metrics used in the article provide a comprehensive assessment of the quality of image descriptions, or are additional metrics needed?

4. How explanatory is the model in generating descriptions? Is it possible to understand why the model generates particular descriptions?

5. Does the article consider the biases that may be introduced by the model and how can these be reduced or eliminated?

6. How useful and accurate are the image descriptions generated by the IT framework in practice?

**Opportunities For Improvement:**

Weaknesses:

1. The process of generating high-quality image descriptions can be computationally intensive, which may be a limitation for researchers or organisations with limited resources. Also the article does not provide a specific analysis of the computational cost to assess its cost-effectiveness in practical applications.

2. The IT framework performs well on specific datasets, but its ability to generalise to image datasets of different types or domains has not yet been verified, and further experiments and validation are needed to see whether the IT framework can be equally effective for domain-specific image description generation.

3. Although several assessment benchmarks have been proposed, these metrics may not be able to comprehensively cover the quality of all types of image descriptions, and some of the assessment metrics may have a certain degree of subjectivity, which is a challenge to further ensure the objectivity and fairness of the assessment results.

4. Image description generation techniques may be used for improper purposes, such as creating false news or misleading information, and such potential risks are not discussed in the article, while issues such as privacy and bias that may be involved in the generation of image descriptions are not explored in depth in the article.

**Relation To Prior Work:**

yes

**Summary And Contributions:**

This paper presents an innovative framework called Image Textualization (IT) that aims to automatically generate high-quality, detailed image descriptions. This framework leverages existing Multimodal Large Language Models (MLLMs) and multiple visual expert models to maximise the conversion of visual information into text in a collaborative manner. The article experimentally validates the effectiveness of the framework and demonstrates that MLLMs trained using the IT framework have significantly improved their ability to generate richer image descriptions

---

> ### Author Rebuttal · Authors · 2024-08-15
>
> Dear Reviewer **g394**,
>
> We are sincerely thankful for your constructive comments and insightful feedback. It is truly encouraging to know that you find our work
> - **1) proposes an innovative framework that solves the challenges in image description**,
> - **2) proposes a comprehensive evaluation covering multiple dimensions of description quality**
> - **3) the generated image descriptions not only contain more details, but also suffer less from hallucination**.
>
> In the following responses, we wish to clarify and address the potential concerns you helped us identify.
>
> **(1) Computational cost**: Our method requires no model tuning and only performance inference, making it computationally efficient. Below is the computational cost for generating 10,000 image descriptions on 8 A40 GPUs with 40G memory:
>
> | Phase 1 | Phase 2 | Phase 3 |
> |:-------:|:-------:|:-------:|
> | 0.65 h  | 0.15 h  | 4.8 h   |
>
> The main computation cost arises from inferencing LLM using the LLaMA 70B model for textualized recaptioning. However, this cost is significantly reduced by acceleration packages like vLLM [1] and can be further optimized using APIs from closed-source LLMs such as GPT-4 [2] or Claude if budget permits. Besides, as there are more efforts focusing on making the LLM more lightweight and compact, we believe in the future, LLMs that are both more powerful and efficient can be incorporated into our framework.
>
> **(2) Different types of image datasets**: Our pipeline primarily targets at daily images with common objects, leveraging vision experts like GRIT[3] or RAM[4]. For domain-specific images, we recommend integrating domain vision experts, such as medical SAM for medical imaging, which fits well within our framework. This will be explored in future work.
>
> **(3) Evaluation metrics**: To evaluate our generated image descriptions, we focus on three main aspects: 1) using established metrics, 2) training MLLMs, and 3) analyzing diffusion model-generated images.
>
> **Aspect 1: Evaluation Using Established Metrics**
>
> We employ various metrics to assess our image descriptions:
>
> - **BLEU**: Assesses n-gram precision.
> - **ROUGE-L**: Measures sequence similarity via the longest common subsequence.
> - **METEOR**: Accounts for precision, recall, and synonym matching.
> - **SPICE**: Evaluates semantic content using scene graphs.
> - **WMD**: Measures semantic similarity with word embeddings.
>
> Using these metrics, we ensure comprehensive evaluation covering precision, recall, sequence structure, and semantics.
>
> **Aspect 2: Training MLLMs**
>
> Our generated descriptions reduce object hallucination in MLLMs, as shown by the object hallucination benchmark.
>
> **Aspect 3: Quality of Diffusion Model-Generated Images**
>
> We use CLIP and DINO-V2 to measure feature similarity between generated and ground truth images. Our descriptions help the diffusion model capture more intricate details.
>
> In conclusion, our evaluation method offers a comprehensive evaluation of the image descriptions. However, we agree that a perfect evaluation for detailed image description still remains challenging, which promotes further research.
>
>
>
> **(4) Is it possible to understand why the model generates particular descriptions?**: Thank you for your suggestion. Examples of our generated descriptions are in the appendix (Figure 1, Tables 6-8). These results show that in the final textualized recaptioning stage, the LLM accurately interprets images and inserts missing objects appropriately, using textual information from MLLM-generated descriptions and spatial information derived from vision expert models. We provide detailed prompts and in-context examples in the appendix and our codebase for reproduction.
>
>
> **(5) Model induced bias**: Thank you for the suggestion. We addressed model-induced bias by using different MLLMs (GPT-4V, LLaVA, InternVL) to generate template descriptions in the first phase. Our framework supports easy adaptation of additional MLLMs to diversify the dataset and mitigate bias from a single model. Specific biases (e.g., emotional tone, style) can be removed explicitly during the third phase (textual recaptioning) by instructing the LLM to eliminate such patterns. For example, we can add the instruction, "Remove the sentimental bias from the image description, such as 'beautiful', 'appealing', 'harmonious'," which successfully removes the sentimental bias, and results in plain and objective descriptions.
>
> **(6) How useful and accurate are the image description**: We experimentally demonstrate the quality of our generated image descriptions from three key aspects:
> - 1) producing richer and more accurate image descriptions (Table 1)
> - 2) reducing hallucinations (Table 2)
> - 3) enabling diffusion models to generate more accurate images (Table 4, Figure 3)
> We believe this evaluation help verify the effectiveness of our proposed data generation framework.
>
> Additionally, we are glad that a recent SOTA MLLM used our curated dataset during training [5], showcasing our dataset's potential in improving the overall performance of a generalist MLLM. We expect our proposed framework and generated dataset to further aid in tasks like diffusion model training and foundational model training, which we will explore in future work.
>
> **(7) Image description generation techniques may be used for improper purposes**: Great suggestion. Our work aims to design an automatic framework for creating detailed and accurate image description dataset to facilitate model training. Although beyond the scope of our work, we agree that the risk of privacy leakage and misleading information are important concerns, which we will clarify in the limitation section of our next version.
>
> [1] Efficient Memory Management for Large Language Model Serving with Paged Attention
>
> [2] GPT-4 Technical Report
>
> [3] GRiT: A Generative Region-to-text Transformer for Object Understanding
>
> [4] Recognize Anything: A Strong Image Tagging Model
>
> [5] LLaVA-OneVision: Easy Visual Task Transfer

---

> ### Author Response · Authors · 2024-08-20
> **Looking forward to further discussion**
>
> Dear Reviewer g394,
>
> Thank you for your constructive feedback and insightful suggestions. Your positive remarks about our work are very encouraging, particularly your recognition that it:
>
> - Proposes an innovative framework addressing the challenges in image description,
> - Presents a comprehensive evaluation covering multiple dimensions of description quality,
> - Generates image descriptions that are more detailed and less prone to hallucination.
>
> We are grateful for your assistance in identifying potential concerns in our manuscript, which we have diligently addressed in our response. With the discussion period deadline approaching, we would greatly appreciate any further comments you may have regarding our response, as we are eager to address any additional issues.
>
> If you find that our responses have adequately addressed your concerns, we would really appreciate it if you could consider raising the score.
>
> We fully understand the demands on your time and sincerely thank you for your efforts in helping us improve our work. We look forward to receiving any additional feedback you may have.

---

> ### Author Response · Authors · 2024-08-27
> **Waiting for to further discussion**
>
> Dear Reviewer g394,
>
> Thank you for your constructive feedback and insightful suggestions. Your positive remarks about our work are very encouraging to us. We are also very grateful for your assistance in identifying potential concerns in our manuscript, which help us improve our paper, and we have diligently addressed in our response.
>
> However, we wish to kindly remind you that **the discussion period will end in less than 3 days**, we would greatly appreciate any further comments you may have regarding our response, as we are eager to address any additional issues.
>
> If you find that our responses have adequately addressed your concerns, we would greatly appreciate it if you could consider raising the score.
>
> We fully understand the demands on your time and sincerely thank you for your efforts in helping us improve our work. We look forward to receiving any additional feedback you may have.

---

> > ### Comment · Reviewer_g394 · 2024-09-01
> >
> > good luck, I support the acceptance of this work.

---

> > > ### Author Response · Authors · 2024-09-01
> > > **Response to Reviewer g394**
> > >
> > > Dear Reviewer g394,
> > >
> > > Thank you for your supportive comments and acknowledgment. We hope that we have successfully addressed your concerns. If so, would you kindly consider updating your score?
> > >
> > > Best regards,
> > >
> > > Authors of Paper 595

---

### Official Review · Reviewer_QK76 · 2024-07-24
**Promising image description dataset construction pipeline**

**Rating:** 8
**Confidence:** 4
**Correctness:** Good.
**Clarity:** Good.

**Review:**

The problem of promoting detail and reducing hallucinations is a central challenge in current image-text tasks, so this pipeline can be of value to the field. The evaluations are promising. In my view, the paper could improve on realistically addressing potential issues and shortcomings of the method.

**Strengths:**

- flexible dataset pipeline
- promising downstream task improvements
- well written

**Additional Feedback:**

- Quotation marks are incorrectly formatted.

**Documentation:**

Good, but I would like it if the prompts could be added to the appendix.

**Ethics:**

No concerns.

**Limitations:**

- Limitations are not discussed but seem quite crucial. For instance, the paper presents the framework as being robust against hallucinations and wrong information more generally. However, even the provided examples contain inaccuracies (e.g., in Figure 2, saying "a silver metal cup" even though there are two). The work shows that it leads to downstream performance, which is great evidence for the benefit of the dataset but I think the paper needs to be more careful about potentially overstating achievements.
- Furthermore, using a purely model-based approach also has limitations, especially when the models used for the generation pipeline are not diverse. For instance, GPT-4V has a strong positivity bias when describing images, e.g., it tends to describe a lot of pictures as cozy and beautiful (as is also the case for the example provided in the paper). When you're exclusively training or evaluating on this type of data, the model biases become a central feature of the data itself.

**Opportunities For Improvement:**

- I disagree with the assumption behind the statement that a "high-quality image description should convey the same information as the corresponding image" (line 31), which I believe to be that it is feasible to create a complete description for an image, so that it can be exactly replicated. Conceptually, this means descriptions would have to describe every single blade of grass, every pixel, every potential association the image elicits, etc. This is infeasible and clearly not the goal of the paper either. It seems that the goal is to **increase** the number of detail and coverage of the image in order to promote more faithful downstream task performance. This doesn't require a fundamental change of the paper but reframing it accordingly sets a more realistic goal post for what the paper attempts to achieve.

**Relation To Prior Work:**

Good.

**Summary And Contributions:**

The paper proposes an automatic construction pipeline for a detailed image description dataset. It leverages a VLM to generate a base description, "vision expert" models that verify the presence of central objects and detect additional ones, an LLM that integrates the base description with the vision expert annotations. The resulting descriptions are more detailed and contain less hallucinations using a pipeline that's more scalable than general crowd-sourcing efforts. The authors show that based on their data, models show improved downstream performance on image-to-text, text-to-image, and hallucination detection tasks.

---

> ### Author Rebuttal · Authors · 2024-08-15
>
> Dear Reviewer **QK76**,
>
> We sincerely appreciate your constructive feedback and insightful advice. It is truly encouraging to know that you find our work
> - **1) proposes a flexible dataset pipeline with potential value to the field**
> - **2) demonstrates promising downstream task improvements**
> - **3) the evaluations are promising**.
>
> In the following responses, we wish to address the potential concerns you helped us identify.
>
>
> **(1) High-quality image description should not convey the same information as the corresponding image**: Thank you for raising this point. In the paper, we intended to express that image descriptions should capture essential visual details to help illustrate what the image looks like, instead of replicating it pixel-for-pixel. We agree that such replication would be difficult to achieve. Furthermore, such replication may be disadvantageous due to the redundant details contained in the description. We'll reframe this in our next version.
>
> **(2) Limitations are not discussed**: Thank you for the suggestion. We agree that while our framework's vision experts excel at fine-grained perception, some corner cases remain challenging. Although our framework produces more detailed and accurate descriptions than current approaches, it doesn't ensure perfect accuracy. Nevertheless, we expect even better generation quality upon the advent or more advanced vision experts and (M)LLMs. We'll add this limitation to the paper in the next version.
>
> **(3) Purely Model-Based Approach Also Has Limitations and Model Bias**: Great suggestion! In fact, we incorporated different MLLMs (GPT-4V, LLaVA, InternVL) to generate template descriptions in the first holistic textualization phase. Our framework supports adaptation of additional MLLMs to diversify the dataset and alleviate the bias of a single MLLM.
>
> In addition, during the rebuttal period, we tried directly adding prompts in the final recaptioning phase to guide the LLM in eliminating such bias. Specifically, we added the instruction, "Remove the sentimental bias from the image description, such as 'beautiful', 'appealing', 'harmonious'," which successfully resulted in plain and objective descriptions. We provide an example below:
>
> **Original:** The image **vividly** captures two contrasting bathroom interiors, each with its unique design elements and functional fixtures that reflect distinct aesthetics of comfort and style. On the left side, the bathroom showcases a **soothing atmosphere** with its upper walls painted in a tranquil blue and the lower walls in a refreshing green. The focal point is a light green toilet, **perfectly** blending with the lower wall, and equipped with a light green toilet seat, emphasizing the **cool color palette**. Nearby, a towel on a rack hints at meticulous upkeep and functionality. Additional touches include a box of tissues and a roll of toilet paper, conveniently placed, ensuring everything is within reach ... a silver trash can, subtly positioned near the toilet, further speaks to **the thoughtful arrangement of this space**. Transitioning to the right, the bathroom adopts a warmer and more **vibrant decor** with its **lively** yellow walls adorned with a tree decal, creating an inviting natural ambiance. A white toilet anchors this space, complemented by a dark brown cabinet mounted on the wall, which hosts a clear jar with a lid, jars with white napkins, totally three jars on a shelf, all contributing to the room’s **warm and cozy feel**. A small white towel hangs on a towel rack above a silver metal bathroom trash can, combining practicality with style. A window dressed with white window blinds offers a view of the outdoors, linking the room with nature and allowing natural light to brighten the space. Both bathrooms, through their deliberate use of colors, fixtures, and decorative elements, emphasize cleanliness, orderliness, and a **harmonious** integration with nature. The **thoughtful** placement of everyday items like toiletries on a shelf and a silver trash can ensures that both practicality and aesthetics are beautifully balanced, offering a **comforting and stylish environment**.
>
> **Modified:** The image depicts two contrasting bathroom interiors, each with unique design elements and functional fixtures. On the left side, the bathroom has upper walls painted in blue and lower walls in green. The toilet is light green, matching the lower wall, and has a light green toilet seat. A towel on a rack is nearby. There is a box of tissues and a roll of toilet paper conveniently placed. A red and white bag hangs from a hook. A window in the bathroom provides an outdoor view. A silver trash can is positioned near the toilet. On the right side, the bathroom has yellow walls with a tree decal. The toilet is white and there is a dark brown cabinet mounted on the wall, with a clear jar with a lid, and jars with white napkins on a shelf. A small white towel hangs on a towel rack above a silver metal trash can. A window with white blinds offers a view of the outdoors. Both bathrooms use colors, fixtures, and decorative elements to emphasize cleanliness and orderliness. Everyday items like toiletries on a shelf and a silver trash can are placed practically.
>
>
> We wish to note that the advantage of our automatic framework is its ability to generate high-quality image descriptions at scale, a task prohibitively expensive if done manually. Given that current image generation models and MLLMs need billions of training samples to achieve competitive performances, our method offers a cost-efficient way to curate high-quality data. While it does not solve all the issues yet, we anticipate improvements with the development of more advanced vision experts, MLLMs and LLMs.
>
> **(4) Add prompts to appendix**: We included the prompts and in-context examples in the appendix. We will clarify more in the next version of our paper.
>
> **(5) Quotation format**: Thank you! We will modify those typos in our next version.

---

> ### Author Response · Authors · 2024-08-20
> **Looking forward to further discussion**
>
> Dear Reviewer QK76,
>
> We sincerely appreciate your constructive feedback and insightful advice. It is truly encouraging to know that you find our work:
>
> - Proposes a flexible dataset pipeline with potential value to the field
> - Demonstrates promising improvements in downstream tasks
> - Offers promising evaluations
>
> We are grateful for your assistance in identifying potential concerns in our manuscript, which we have diligently addressed in our response. With the discussion period deadline approaching, we would greatly appreciate any further comments you may have on our response, as we are eager to address any additional concerns. If you find our responses to have addressed your concerns, we would appreciate it if you could consider raising the score.
>
> We fully understand your busy schedule and are genuinely thankful for the time you have dedicated to helping us enhance our work. We look forward to receiving any additional feedback you may have.

---

> > ### Comment · Reviewer_QK76 · 2024-08-26
> >
> > I thank the authors for the clarifications they provided to my questions and comments. They all sound great to me and I increased my score accordingly, assuming that the suggestions will be incorporated. I do want to encourage the authors to discuss the potential biases induced by models in the paper. While it's a great experiment to see that you can reduce the "flowery" language, it is only an example of the biases that might be induced without us being aware of it. I like the point of using a suite of models to diversify but I do believe it's still important to flag the bias issue.

---

> > ### Author Response · Authors · 2024-08-27
> > **Response to Reviewer QK76**
> >
> > Dear Reviewer QK76,
> >
> > Thank you very much for your response! We are really glad you find that our previous responses addressed some of the concerns.
> >
> > We agree that potential biases induced by the MLLM during the generation of image descriptions remains to be challenging. Although utilizing a suite of MLLMs might help alleviate this issue, completely addressing all biases is difficult, as some may not be easily detectable by humans. We will incorporate this limitation into our updated version.
> >
> > Once again, we appreciate your insightful suggestions and valuable advice, which have greatly helped improve our paper. We are also very pleased to see the score increase.

---

### Official Review · Reviewer_aJpW · 2024-08-04
**Good paper with some opportunities to improve**

**Rating:** 6
**Confidence:** 4
**Clarity:** Yes.

**Review:**

Overall, the paper is clear, sound, original, and helpful for the research community.

Please refer to the below sections for detailed comments.

**Strengths:**

Sentences are easy to read.

The procedure is straightforward to understand.

The problem statement is clear.

The solution almost corresponds to the problem statement.

**Additional Feedback:**

I am willing to raise my score upon improvements.

**Correctness:**

I am not sure if 200 samples is enough for evaluation.

CLIP is not a proper metric for D2I because it is trained on the pairs of images and non-detailed texts.

The experiments do not report the length and detailedness of IT-finetuned LLaVA-7B explicitly.

**Documentation:**

The paper mostly focuses on language part and misses details in collecting images.

**Ethics:**

No.

**Limitations:**

Limitations in the appendix covers minimal future work: doing the same thing with LLaVA-70B.

**Opportunities For Improvement:**

Adding a flowchart of the dataset generating procedure would greatly help the readers to understand it.

What would be the effect of IT-curated descriptions on stable diffusion?

Example descriptions in the existing web-based or annotated datasets would support the problem statement.

The source of the images should be mentioned.

L17 should note 'fine-tuning', instead of simply 'training' on IT-curated descriptions.

Typos
* L79 rich viual details -> rich visual details

**Relation To Prior Work:**

Yes.

**Summary And Contributions:**

This paper introduces a dataset of image-description pairs.

Problem statement:
* Image-text pairs from the web are low quality and noisy.
* COCO descriptions are short and coarse.
* Human annotations are expensive.

Motivation
* Multi-modal LLMs often suffer from lack of details and hallucination.
* Vision models are trained with high-res images and fine-grained object-level annotations.

Proposed framework: Image textualization
* Its procedure is ***automatic*** with below procedure.
  * reference description := MLLM(image)
  * object entities := LLM(reference description, "extract entities")
  * hallucination tags := Whether GroundingDino(object entities) exists or not
  * fine-grained object-level attributes and bboxes := DenseCaptioner(image)
  * depthmap := MonocularDepthEstimation(image)
  * object depths := Average depthmap for each mask in SAM(image) $\cap$ bboxes
  * textualized recaption := LLM(fine-grained object-level attributes, object depths)
* The descriptions are high quality (in what aspect?)

Proposed benchmarks for image captioning
* Targets: Captioners
* Detailed Image Description (DID)
  * reference descriptions := LLaVA(100 images) $\cup$ GPT4-V(100 images)
  * GT descriptions := manually add missing details and remove hallucinations(reference descriptions)
  * for captioner in Captioners:
    * for metric in [BLEU, ROUGE-L, METEOR, SPICE, WMD]:
      * metric(GT descriptions, captioner(200 images))
* Description-to-Image (D2I)
  * for captioner in Captioners:
    * captions := captioner(orignal images)
    * generated images := PixArt(captions)
    * similarity := Cosine similarity(CLIP(original images), CLIP(generated images)
* Linguistic (LIN)
  * for captioner in Captioners:
    * for metric in [ARI, FK, SMOG]
      * metric(captioner(image))
* POPE [31]

LLaVA-7B fine-tuned on IT-curated descriptions provides richer captioning capability regarding
* longer
* detailed
* less hallucination

Statistical analysis

---

> ### Author Rebuttal · Authors · 2024-08-15
>
> Dear Reviewer **aJpW**,
>
> We deeply appreciate your insightful suggestions and helpful comments. We are glad and honored to see that you acknowledge our work to be **1) clear, sound, original, and helpful for the research community**, **2) procedure is straightforward to understand**, and **3) The problem statement is clear.**  We aim to address the potential concerns you have raised in our paper in the following responses:
>
> **(1) Flowchart**: Great suggestion! In figure 1 of the original paper, we drew a diagram to illustrate our data generation framework. However, we highly agree that including a flow chart may present the pipeline more clearly and formally. We have now included a flowchart in the submitted PDF and will update it in the next paper version.
>
> **(2) IT-curated descriptions on stable diffusion**: The current open-source stable diffusion model supports input text up to 100 words due to text encoder limitations. We chose PixArt[1] (which supports 300 words) to evaluate our generated descriptions for text-to-image generation. Increasing the text encoder's input length would allow our detailed IT descriptions, which include precise visual details, to enhance model performance and offer more accurate supervision during training.
>
> **(3) Example web-scraped descriptions**: Great suggestion! We visualized a few web images and their descriptions in the PDF. These descriptions are often short, incomplete, and noisy. We will include examples from multiple data sources in our revised version.
>
> **(4) Image source**: Thank you for the reminder! We will add the sources of the images in our paper.
>
> **(5) Fine-tuning -> training & Typo**: We apologize for the careless mistakes and will fix them in the next version.
>
> **(6) Limitation & future work**: Thank you for your suggestion. Our paper primarily focuses on creating image description datasets containing daily objects, which is the most common scenario. However, studying domain-specific data, such as medical imaging, is also an important and practical direction. We believe involving domain-specific vision experts can help identify novel objects in these images. We will incorporate this future work in our updated version.
>
> **(7) Number of evaluation samples**: For evaluation, we ensure benchmark accuracy by manually verifying each of the 200 samples, a size similar to MMvet[2] and LLaVA-wild[3]. While our curated evaluation benchmarks are important to verify the data quality, we wish to note that our primary contribution is the design of a scalable and automatic framework for creating high-quality image descriptions. Recently, we are pleased to see that a SOTA MLLM adopted our curated dataset for training[4], which proves its quality and effectiveness for improving the model's overall performance.
>
> **(8) CLIP is not a proper metric**: Although CLIP is trained on image-text pairs with non-detailed texts, its substantial pretraining data allows the vision encoder to capture rich semantic features from images. This is partially why concurrent MLLMs adopt CLIP-pretrained vision encoders. If the CLIP features of the generated and original images are highly similar, it suggests similar semantics between the images.
>
> However, we agree that CLIP features may lack fine-grained visual information due to the global image-caption contrastive learning objective. Thus, we also used the DINO-V2[5] encoder for evaluating feature similarity, as it is trained using self-supervised objective and captures richer visual details[6]. The D2I-Bench results are provided in Table 4 of the original paper, we also provide the results below and in the rebuttal PDF for clarity. From the table below, we observe that IT-generated descriptions enable the generation of synthetic images that lead to both higher CLIP score and DINOV2 score.
>
> | Description | CLIP score | DINOV2 Score |
> |-----------------|----------------|----------------------|
> | COCO | 72.24 | 77.84 |
> | {LLaVA} | 73.44 | 80.39 |
> | IT-{LLaVA} | **74.27** | **81.20** |
> | {GPT-4V} | 76.49 | 82.80 |
> | IT-{GPT-4V} | **77.10** | **83.71** |
>
> **(9) The length and detailedness of IT-finetuned LLaVA-7B**: Great suggestion! We compare the lengths of descriptions generated from LLaVA-7B tuned with MLLM-Generated descriptions and IT-generated descriptions respectively in the table below and the provided PDF:
> | Model | # of Average Sentences| # of Average Words |
> |-----------------|----------------|----------------------|
> | LLaVA-7B-MLLM | 8.68 | 152.13 |
> | LLaVA-7B-IT| **10.21** | **190.07** |
>
> After tuning with our generated image descriptions, LLaVA-7B is able to produce more detailed descriptions with more sentences and words.
>
> **(10) Misses details in collecting images**: Thank you for the reminder. In this paper, we focus on creating high-quality image descriptions from the language part, as images are much easier to obtain. For instance, datasets like LAION[7] and COYO[8] collect millions or even billions of images, but the major issue is the poor quality of textual descriptions. In our work, we simply leverage images from ShareGPT4V[9] dataset. We agree that we need to emphasize this point in our paper.
>
> [1] PixArt-α: Fast Training of Diffusion Transformer for Photorealistic Text-to-Image Synthesis
>
> [2] MM-Vet: Evaluating Large Multimodal Models for Integrated Capabilities
>
> [3] Visual Instruction Tuning
>
> [4] LLaVA-OneVision: Easy Visual Task Transfer
>
> [5] DINOv2: Learning Robust Visual Features without Supervision
>
> [6] Eyes Wide Shut? Exploring the Visual Shortcomings of Multimodal LLMs
>
> [7] LAION-5B: An open large-scale dataset for training next generation image-text models
>
> [8] COYO-700M: Large-scale Image-Text Pair Dataset
>
> [9] Improving Large Multi-Modal Models with Better Captions

---

> ### Author Response · Authors · 2024-08-25
> **Looking forward to further discussion**
>
> Dear Reviewer aJpW,
>
> We deeply appreciate your insightful suggestions and helpful comments. It is an honor to know that you find our work clear, sound, original, and beneficial to the research community. We are also pleased to hear that you recognize the straightforward nature of our procedure and the clarity of our problem statement.
>
> Your help in identifying potential concerns in our manuscript has been invaluable, and we have diligently addressed these in our response. Since the discussion period deadline is approaching, we would greatly appreciate any further comments you may have on our response. We are eager to resolve any additional concerns you might have.
>
> We fully understand your busy schedule and are genuinely thankful for the time you have dedicated to helping us enhance our work. We look forward to receiving any additional feedback you may offer.

---

> ### Author Response · Authors · 2024-08-27
> **Waiting for to further discussion**
>
> Dear Reviewer aJpW,
>
> Thank you so much for your constructive feedback and insightful suggestions. Your positive remarks about our work are very encouraging. We are grateful for your assistance in identifying potential concerns in our manuscript, which help us improve our paper, and we have diligently addressed in our response.
>
> However, we wish to kindly remind you that **the discussion period will end in less than 3 days**, we would greatly appreciate any further comments you may have regarding our response, as we are eager to address any additional issues.
>
> If you find that our responses have adequately addressed your concerns, we would greatly appreciate it if you could consider raising the score.
>
> We fully understand the demands on your time and sincerely thank you for your efforts in helping us improve our work. We look forward to receiving any additional feedback you may have.

---

> ### Author Response · Authors · 2024-08-31
> **Waiting for to further discussion**
>
> Dear Reviewer **aJpW**,
>
> Thank you for your constructive feedback and insightful suggestions. Your positive remarks about our work are truly encouraging. We are grateful for your assistance in identifying potential concerns in our manuscript, which have helped us improve our paper, and we have diligently addressed these in our response.
>
> However, **with the discussion period coming to a close in less than a day**, we kindly request any further comments you may have regarding our response. We are eager to address any additional issues that you might identify.
>
> If you believe our responses have satisfactorily addressed your concerns, we would be most appreciative if you could consider raising the score.
>
> We fully understand the demands on your time and sincerely thank you for your efforts in helping us improve our work. We look forward to any additional feedback you may have.

---

> > ### Comment · Reviewer_aJpW · 2024-09-01
> >
> > I appreciate the effort of the authors in the rebuttal.
> >
> > Most of my concerns are resolved except (5).
> >
> > The paper became stronger after the rebuttal.
> >
> > (I do not find the [edit] button in my review.)
> >
> > I keep my positive rating.

---

> > > ### Author Response · Authors · 2024-09-01
> > > **Response to Reviewer aJpW**
> > >
> > > Dear Reviewer aJpW,
> > >
> > > Thank you very much for your acknowledgement! We really appreciate your time and effort in helping us improve our paper. We will update the typos in our final version of the paper.
> > >
> > > Meanwhile, as the rating 6 is "marginally above acceptance threshold", we would really appreciate it if you could consider raising the score if you find our responses to have addressed the concerns.
> > >
> > > Best Regards,
> > >
> > > Authors of submission 595

---

### Author Rebuttal · Authors · 2024-08-19

Dear Reviewers,

We sincerely appreciate the time and effort you have invested in reviewing our manuscript and providing such insightful suggestions and comments. We are pleased that you find our paper:

- Proposes a useful and practical dataset pipeline (aJpW, QK76, g394, MZYA).
- Demonstrates promising performance (QK76, g394).
- Provides a comprehensive evaluation (QK76, g394).

Your feedback has been invaluable in helping us improve our manuscript. In this rebuttal, we have made our best effort to address the concerns you raised. However, as the deadline for the discussion period is approaching, we would greatly appreciate any further comments you may have on our response. We are eager to address any additional concerns you might have.

We fully understand how busy you are, and we are truly grateful for the time you have dedicated to helping us enhance our work. We look forward to receiving any additional feedback you may have.

---

### Author Response · Authors · 2024-08-26
**Waiting for further discussion**

Dear Reviewers,

We sincerely appreciate the time and effort you have invested in reviewing our manuscript and providing such insightful suggestions and comments. Your feedback has been invaluable in helping us improve our work. In this rebuttal, we have made our best effort to address the concerns you raised.

However, since **there are only 2 days away from the end of discussion period**, we humbly seek your further comments on our response. We fully understand how busy you are, and we are truly grateful for your dedication. We are eager to address any additional concerns you might have.

Thank you for your attention.

---

### Author Response · Authors · 2024-08-31
**Waiting for further discussion with the reviewers**

Dear Reviewers,

We sincerely appreciate the time and effort you have invested in reviewing our manuscript and providing such insightful suggestions and comments. We are also thrilled to know that the acknowledged that our paper:

- Proposes a useful and practical dataset pipeline (aJpW, QK76, g394, MZYA).
- Demonstrates promising performance (QK76, g394).
- Provides a comprehensive evaluation (QK76, g394).

Your feedback has been invaluable in helping us improve our work. In this rebuttal, we have made our best effort to address the concerns you raised.

However, **with only one day remaining in the discussion period**, we humbly seek your further comments on our response.

We fully understand how busy you are, and we are truly grateful for your dedication. We are eager to address any additional concerns you might have.

Thank you for your attention.

Best regards,

Authors of paper 595

---

### Decision · Program_Chairs · 2024-09-26

**Decision:**

Accept (Poster)

**Comment:**

The paper has received unanimous approval from all the reviewers and deals with a interesting and important concept of text - to - image and image-to-text generation with high fidelity. I recommend an approval as a poster.